# Risk-Averse Bayes-Adaptive Reinforcement Learning

**Marc Rigter**
Oxford Robotics Institute
University of Oxford
mrigter@robots.ox.ac.uk

**Bruno Lacerda**
Oxford Robotics Institute
University of Oxford
bruno@robots.ox.ac.uk

**Nick Hawes**
Oxford Robotics Institute
University of Oxford
nickh@robots.ox.ac.uk

## Abstract

In this work, we address risk-averse Bayes-adaptive reinforcement learning. We pose the problem of optimising the conditional value at risk (CVaR) of the total return in Bayes-adaptive Markov decision processes (MDPs). We show that a policy optimising CVaR in this setting is risk-averse to both the epistemic uncertainty due to the prior distribution over MDPs, and the aleatoric uncertainty due to the inherent stochasticity of MDPs. We reformulate the problem as a two-player stochastic game and propose an approximate algorithm based on Monte Carlo tree search and Bayesian optimisation. Our experiments demonstrate that our approach significantly outperforms baseline approaches for this problem.

## 1 Introduction

In standard model-based reinforcement learning (RL), an agent interacts with an unknown environment to maximise the expected reward [42]. However, for any given episode the total reward received by the agent is uncertain. There are two sources of this uncertainty: the *epistemic* (or *parametric*) uncertainty due to imperfect knowledge about the underlying MDP model, and *aleatoric* (or *internal*) uncertainty due to the inherent stochasticity of the underlying MDP [20]. In many domains, we wish to find policies which are risk-averse to both sources of uncertainty.

As an illustrative example, consider a navigation system for an automated taxi service. The system attempts to minimise travel duration subject to uncertainty due to the traffic conditions, and traffic lights and pedestrian crossings. For each new day of operation, the traffic conditions are initially unknown. This corresponds to epistemic uncertainty over the MDP model. This uncertainty is reduced as the agent collects experience and learns which roads are busy or not busy. Traffic lights and pedestrian crossings cause stochasticity in the transition durations corresponding to aleatoric uncertainty. The aleatoric uncertainty remains even as the agent learns about the environment. A direct route, i.e. through the centre of the city, optimises the expected journey duration. However, occasionally this route incurs extremely long durations due to some combination of poor traffic conditions and getting stuck at traffic lights and pedestrian crossings. Thus, bad outcomes can be caused by a combination of both sources of uncertainty. These rare outcomes may result in unhappy customers for the taxi service. Therefore, we wish to be risk-averse and prioritise greater certainty of avoiding poor outcomes rather than expected performance. To ensure that the risk of a bad journey is avoided, the navigation system must consider both the epistemic and the aleatoric uncertainties.

In model-based Bayesian RL, a belief distribution over the underlying MDP is maintained. This quantifies the epistemic uncertainty over the underlying MDP given the transitions observed so far.

35th Conference on Neural Information Processing Systems (NeurIPS 2021).

The Bayesian RL problem can be reformulated as a planning problem in a *Bayes-Adaptive* MDP (BAMDP) with an augmented state space composed of the belief over the underlying MDP and the state in the underlying MDP [12].

In this work, we address risk-averse decision making in model-based Bayesian RL. Instead of optimising for expected value, we optimise a risk metric applied to the total return of the BAMDP. Specifically, we focus on *conditional value at risk* (CVaR) [34]. The return in a BAMDP is uncertain due to both the uncertain prior belief over the underlying MDP, and the inherent stochasticity of MDPs. By optimising the CVaR of the return in the BAMDP, our approach simultaneously addresses epistemic *and* aleatoric uncertainty under a single framework. This is in contrast with previous works which have generally considered risk-aversion to either epistemic *or* aleatoric uncertainty.

We formulate CVaR optimisation in Bayesian RL as a stochastic game over the BAMDP against an adversarial environment. Solving this game is computationally challenging because the set of reachable augmented states grows exponentially, and the action space of the adversary is continuous. Our proposed algorithm uses Monte Carlo tree search (MCTS) to focus search on promising areas of the augmented state space. To deal with the continuous action space of the adversary, we use Bayesian optimisation to expand promising adversary actions. Our main contributions are:

- Addressing CVaR optimisation of the return in model-based Bayesian RL to simultaneously mitigate epistemic and aleatoric uncertainty.
- An algorithm based on MCTS and Bayesian optimisation for this problem.

To our knowledge, this is the first work to optimise a risk metric in model-based Bayesian RL to simultaneously address both epistemic and aleatoric uncertainty, and the first work to present an MCTS algorithm for CVaR optimisation in sequential decision making problems. Our empirical results show that our algorithm significantly outperforms baseline methods on two domains.

## 2  Related Work

Many existing works address aversion to risk stemming from either epistemic model uncertainty or aleatoric uncertainty. One of the most popular frameworks for addressing epistemic uncertainty is the Robust or Uncertain MDP [17, 26, 33, 47] which considers worst-case expected value over a set of possible MDPs. Other works address epistemic uncertainty by assuming that a distribution over the underlying MDP is known. One approach is to find a policy which optimises the expected value in the worst k% of parameter realisations [5, 11]. Most related to our work is that of Sharma et al. [38] which optimises a risk metric in the Bayesian RL context where the prior is a discrete distribution over a finite number of MDPs. However, all of these existing works marginalise out the aleatoric uncertainty by considering the expected value under each possible MDP. In contrast, we optimise CVaR of the return in the Bayesian RL setting to address both epistemic and aleatoric uncertainty.

Conditional value at risk (CVaR) [34] is a common coherent risk metric which has been applied to MDPs to address aleatoric uncertainty. For an introduction to coherent risk metrics, see [2]. In this paper, we consider *static* risk, where the risk metric is applied to the total return, rather than *dynamic* risk, where the risk metric is applied recursively [43]. For this static CVaR setting, approaches based on value iteration over an augmented state space have been proposed [3, 8, 32]. Other works propose policy gradient methods to find a locally optimal solution for the optimisation of CVaR [4, 6, 7, 31, 43, 45, 46], or general coherent risk metrics [44]. These existing approaches all assume the agent has access to the true underlying MDP for training/planning, and optimise the CVaR in that single MDP. In our work, we address the model-based Bayesian RL setting where we only have access to a prior belief distribution over MDPs. We find Bayes-adaptive policies which are risk-averse to both the epistemic uncertainty over the underlying MDP and the aleatoric uncertainty due to stochastic transitions. To our knowledge, no existing work has addressed CVaR optimisation in the Bayes-adaptive RL setting to simultaneously address both forms of uncertainty.

Constrained MDPs impose a constraint on the expected value of a cost [1, 36]. Such constraints have been addressed in Bayesian RL [18]. However, this approach only constrains the expected cost and it is unclear how to choose cost penalties to obtain the desired risk averse behaviour. Monte Carlo Tree Search (MCTS) has been adapted to expected cost constraints [19]. However, to our knowledge, this is the first work to adapt MCTS to CVaR optimisation in sequential decision making problems.

In the Deep RL setting, Deep Q-Networks [24] have been modified to predict the uncertainty over the Q-values associated with epistemic and aleatoric uncertainty [9, 13] and derive a risk-sensitive policy. In this work, we propose to use the Bayes-adaptive formulation of RL to model and address both uncertainty sources. Another related strand of work is robust adversarial reinforcement learning [28, 30] (RARL) which optimises robust policies by simultaneously training an adversary which perturbs the environment. In RARL it is unclear how much power to give to the adversary to achieve the desired level of robustness. Our approach ensures that the CVaR at the desired confidence level is optimised.

## 3  Preliminaries

Let $Z$ be a bounded-mean random variable, i.e. $E[|Z|] < \infty$, on a probability space $(\Omega, \mathcal{F}, \mathcal{P})$, with cumulative distribution function $F(z) = \mathcal{P}(Z \leq z)$. In this paper we interpret $Z$ as the total reward, or return, to be maximised. The *value at risk* at confidence level $\alpha \in (0, 1]$ is defined as $\text{VaR}_\alpha(Z) = \min\{z | F(z) \geq \alpha\}$. The *conditional value at risk* at confidence level $\alpha$ is defined as

$$\text{CVaR}_\alpha(Z) = \frac{1}{\alpha} \int_0^\alpha \text{VaR}_\gamma(Z) d\gamma. \tag{1}$$

If $Z$ has a continuous distribution, $\text{CVaR}_\alpha(Z)$ can be defined using the more intuitive expression: $\text{CVaR}_\alpha(Z) = \mathbb{E}[Z | Z \leq \text{VaR}_\alpha(Z)]$. Thus, $\text{CVaR}_\alpha(Z)$ may be interpreted as the expected value of the $\alpha$-portion of the left tail of the distribution of $Z$. CVaR may also be defined as the expected value under a perturbed distribution using its dual representation [34, 37]

$$\text{CVaR}_\alpha(Z) = \min_{\xi \in \mathcal{B}(\alpha, \mathcal{P})} \mathbb{E}_\xi [Z], \tag{2}$$

where $\mathbb{E}_\xi[Z]$ denotes the $\xi$-weighted expectation of $Z$, and the *risk envelope*, $\mathcal{B}$, is given by

$$\mathcal{B}(\alpha, \mathcal{P}) = \left\{ \xi : \xi(\omega) \in \left[0, \frac{1}{\alpha}\right], \int_{\omega \in \Omega} \xi(\omega) \mathcal{P}(\omega) d\omega = 1 \right\}. \tag{3}$$

Therefore, the CVaR of a random variable $Z$ may be interpreted as the expectation of $Z$ under a worst-case perturbed distribution, $\xi \mathcal{P}$. The risk envelope is defined so that the probability density of any outcome can be increased by a factor of at most $1/\alpha$, whilst ensuring the perturbed distribution is a valid probability distribution.

### 3.1  CVaR Optimisation in Standard MDPs

A finite-horizon Markov decision process (MDP) is a tuple, $\mathcal{M} = (S, A, R, T, H, s_0)$, where $S$ and $A$ are finite state and action spaces, $R : S \times A \to \mathbb{R}$ is the reward function, $T : S \times A \times S \to [0, 1]$ is the transition function, $H$ is the horizon, and $s_0$ is the initial state. A history of $\mathcal{M}$ is a sequence $h = s_0 a_0 s_1 a_1 \ldots a_{t-1} s_t$ such that $T(s_i, a_i, s_{i+1}) > 0$ for all $i$. We denote the set of all histories over $\mathcal{M}$ as $\mathcal{H}^\mathcal{M}$. A history-dependent policy is a function $\pi : \mathcal{H}^\mathcal{M} \to A$, and we write $\Pi_\mathcal{H}^\mathcal{M}$ to denote the set of all history-dependent policies for $\mathcal{M}$. If $\pi$ only depends on the last state $s_t$ of $h$, then we say $\pi$ is *Markovian*, and we denote the set of all Markovian policies as $\Pi^\mathcal{M}$. A policy $\pi$ induces a distribution $\mathcal{P}_\pi^\mathcal{M}$ over histories and we define $G_\pi^\mathcal{M} = \sum_{t=0}^H R(s_t, a_t)$ as the distribution over total returns of $\mathcal{M}$ under $\pi$. Many existing works have addressed the problem of optimising the static CVaR of $G_\pi^\mathcal{M}$, defined as follows.

**Problem 1 (CVaR optimisation in standard MDPs)** *Let $\mathcal{M}$ be an MDP. Find the optimal CVaR of the total return at confidence level $\alpha$:*

$$\max_{\pi \in \Pi_\mathcal{H}^\mathcal{M}} \text{CVaR}_\alpha(G_\pi^\mathcal{M}). \tag{4}$$

Unlike dynamic Markov risk measures [35], history-dependent policies may be required to optimally solve this static CVaR optimisation problem [3, 8, 43]. Previous works [8, 27] have shown that this

problem can be interpreted either as being risk-averse to aleatoric uncertainty, *or* as being robust to epistemic uncertainty. The latter interpretation comes from the dual representation of CVaR (Eq. 2), where CVaR is expressed as the expected value under a worst-case perturbed probability distribution.

Methods based on dynamic programming have been proposed to solve Problem 1 [3, 8, 29]. In particular, Chow et al. [8] augment the state space by an additional continuous state factor, $y \in (0, 1]$, corresponding to the confidence level. The value function for the augmented state $(s, y)$ is defined by $V(s, y) = \max_{\pi \in \Pi_{\mathcal{H}}^{\mathcal{M}}} \text{CVaR}_y(G_\pi^{\mathcal{M}})$, and can be computed using the following Bellman equation

$$V(s, y) = \max_{a \in A} \left[ R(s, a) + \min_{\xi \in \mathcal{B}(y, T(s,a,\cdot))} \sum_{s' \in S} \xi(s') V(s', y\xi(s')) T(s, a, s') \right], \qquad (5)$$

where $\xi$ represents an adversarial perturbation to the transition probabilities, and the additional state variable, $y$, ensures that the perturbation to the probability of any history through the MDP is at most $1/y$. Thus, $y$ can be thought of as keeping track of the "budget" to adversarially perturb the probabilities. Therefore, $V(s_0, \alpha)$ is the expected value under the adversarial probability distribution defined by Eq. 2 and 3 and corresponds to the optimal CVaR in the MDP (Problem 1). To address the continuous state variable, $y$, [8] use dynamic programming with linear function approximation.

### 3.2 Stochastic Games

In this paper, we will formulate CVaR optimisation as a turn-based two-player zero-sum stochastic game (SG). An SG between an agent and an adversary is a generalisation of an MDP and can be defined using a similar tuple $\mathcal{G} = (S, A, T, R, H, s_0)$. The elements of $\mathcal{G}$ are interpreted as with MDPs, but extra structure is added. In particular, $S$ is partitioned into a set of agent states $S^{agt}$, and a set of adversary states $S^{adv}$. Similarly, $A$ is partitioned into a set of agent actions $A^{agt}$, and a set of adversary actions $A^{adv}$. The transition function is defined such that agent actions can only be executed in agent states, and adversary actions can only be executed in adversary states.

We denote the set of Markovian agent policies mapping agent states to agent actions as $\Pi^{\mathcal{G}}$ and the set of Markovian adversary policies, defined similarly, as $\Sigma^{\mathcal{G}}$. A pair $(\pi, \sigma)$ of agent-adversary policies induces a probability distribution over histories and we define $G_{(\pi,\sigma)}^{\mathcal{G}} = \sum_{t=0}^{H} R(s_t, a_t)$ as the distribution over total returns of $\mathcal{G}$ under $\pi$ and $\sigma$. In a zero-sum SG, the agent seeks to maximise the expected return reward, whilst the adversary seeks to minimise it:

$$\max_{\pi \in \Pi^{\mathcal{G}}} \min_{\sigma \in \Sigma^{\mathcal{G}}} \mathbb{E}\left[ G_{(\pi,\sigma)}^{\mathcal{G}} \right]. \qquad (6)$$

### 3.3 Bayesian RL

Here we describe the Bayesian formulation of decision making in an unknown MDP [21, 12]. In Bayesian RL, the dynamics of the MDP $\mathcal{M}$ are unknown and we assume that its transition function, $T$, is a latent variable distributed according to a prior distribution $\mathcal{P}(T|h_0)$, where $h_0$ represents the empty history where no transitions have been observed. After observing a history $h = s_0 a_0 s_1 a_1 \ldots a_{t-1} s_t$, the posterior belief over $T$ is updated using Bayes' rule: $\mathcal{P}(T|h) \propto \mathcal{P}(h|T)\mathcal{P}(T|h_0)$.

In the Bayes-Adaptive MDP (BAMDP) formulation of Bayesian RL [12], the uncertainty about the model dynamics is captured by augmenting the state space to include the history: $S^+ = S \times \mathcal{H}$. This captures the model uncertainty as the history is a sufficient statistic for the posterior belief over $T$ given the initial prior belief. The transition and reward functions for this augmented state space are

$$T^+((s, h), a, (s', has')) = \int_T T(s, a, s')\mathcal{P}(T|h)dT, \qquad (7)$$

$$R^+((s, h), a) = R(s, a). \qquad (8)$$

The initial augmented state is $s_0^+ = (s_0, h_0)$. The tuple $\mathcal{M}^+ = (S^+, A, T^+, R^+, H, s_0^+)$ defines the BAMDP. Typically, solutions to the Bayesian RL problem attempt to (approximately) find a

*Bayes-optimal* policy, which maximises the expected sum of rewards under the prior belief over the transition function. In this work, we address risk-averse solutions to the Bayesian RL problem.

# 4 CVaR Optimisation in Bayes-Adaptive RL

Many existing works address CVaR optimisation in standard MDPs. Here, we assume that we only have access to a prior distribution over MDPs, and optimise the CVaR in the corresponding BAMDP.

**Problem 2 ( CVaR optimisation in Bayes-Adaptive MDPs)** *Let $\mathcal{M}^+$ be a Bayes-Adaptive MDP. Find the optimal CVaR at confidence level $\alpha$:*

$$\max_{\pi \in \Pi^{\mathcal{M}^+}} \mathrm{CVaR}_\alpha(G_\pi^{\mathcal{M}^+}). \tag{9}$$

Posing CVaR optimisation over BAMDPs leads to two observations. First, unlike Problem 1, we do not need to consider history-dependent policies as the history is already embedded in the BAMDP state. Second, the solution to this problem corresponds to a policy which is risk-averse to both the epistemic uncertainty due to the uncertain belief over the MDP, and the aleatoric uncertainty due to the stochasticity of the underlying MDP. In the following, we elaborate on this second observation.

## 4.1 Motivation: Robustness to Epistemic and Aleatoric Uncertainty

The solution to Problem 2 is risk-averse to both the epistemic *and* the aleatoric uncertainty. Intuitively, this is because the return distribution in the BAMDP is influenced both by the prior distribution over models as well as stochastic transitions. In this section, we introduce a novel alternative definition of CVaR in BAMDPs to formally illustrate this concept. Specifically, we show that CVaR in a BAMDP is equivalent to the expected value under an adversarial prior distribution over transition functions, and adversarial perturbations to the transition probabilities in all possible transition functions. The perturbation "budget" of the adversary is split multiplicatively between perturbing the probability density of transition functions, and perturbing the transition probabilities along any history. Therefore, the adversary has the power to minimise the return for the agent through a combination of modifying the prior so that "bad" transition functions are more likely (epistemic uncertainty), and making "bad" transitions within any given transition function more likely (aleatoric uncertainty).

We consider an adversarial setting, whereby an adversary is allowed to apply perturbations to modify the prior distribution over the transition function as well as the transition probabilities within each possible transition function, subject to budget constraints. Define $\mathcal{T}$ to be the support of $\mathcal{P}(T|h_0)$, and let $T_p \in \mathcal{T}$ be a plausible transition function. Note that, in general, $\mathcal{T}$ can be a continuous set. Consider a perturbation of the prior distribution over transition functions $\delta : \mathcal{T} \to \mathbb{R}_{\geq 0}$ such that $\int_{T_p} \delta(T_p)\mathcal{P}(T_p|h_0)dT_p = 1$. Additionally, for $T_p \in \mathcal{T}$, consider a perturbation of $T_p$, $\xi_p : S \times A \times S \to \mathbb{R}_{\geq 0}$ such that $\sum_{s' \in S} \xi_p(s, a, s') \cdot T_p(s, a, s') = 1 \ \forall s, a$. We denote the set of all possible perturbations of $T_p$ as $\Xi_p$, and the set of all perturbations over $\mathcal{T}$ as $\Xi = \times_{T_p \in \mathcal{T}} \Xi_p$. For BAMDP $\mathcal{M}^+$, $\delta : \mathcal{T} \to \mathbb{R}_{\geq 0}$, and $\xi \in \Xi$, we define $\mathcal{M}_{\delta,\xi}^+$ as the BAMDP, obtained from $\mathcal{M}^+$, with perturbed prior distribution $\mathcal{P}^\delta(T_p|h_0) = \delta(T_p) \cdot \mathcal{P}(T_p|h_0)$ and perturbed transition functions $T_p^\xi(s, a, s') = \xi_p(s, a, s') \cdot T_p(s, a, s')$. We are now ready to state Prop. 1, which provides an interpretation of CVaR in BAMDPs as expected value under an adversarial prior distribution and adversarial transition probabilities. All propositions are proven in the supplementary material.

**Proposition 1 (CVaR in BAMDPs)** *Let $\mathcal{M}^+$ be a BAMDP. Then:*

$$\mathrm{CVaR}_\alpha(G_\pi^{\mathcal{M}^+}) = \min_{\substack{\delta : \mathcal{T} \to \mathbb{R}_{\geq 0} \\ \xi \in \Xi}} \mathbb{E}\big[G_\pi^{\mathcal{M}_{\delta,\xi}^+}\big], \quad \textbf{\textit{s.t.}} \tag{10}$$

$$0 \leq \delta(T_p)\xi_p(s_0, a_0, s_1)\xi_p(s_1, a_1, s_2) \cdots \xi_p(s_{H\text{-}1}, a_{H\text{-}1}, s_H) \leq \frac{1}{\alpha},$$
$$\forall \, T_p \in \mathcal{T}, \ \forall (s_0, a_0, s_1, a_1, \dots, s_H) \in \mathcal{H}_H.$$

Prop. 1 shows that CVaR can be computed as expected value under an adversarial prior distribution over transition functions, and adversarial perturbations to the transition probabilities in all possible transition functions. Therefore, a policy which optimises the CVaR in a BAMDP must mitigate the risks due to both sources of uncertainty.

## 4.2 Bayes-Adaptive Stochastic Game Formulation

In Prop. 1, we formulated CVaR in BAMDPs as expected value under adversarial perturbations to the prior, and to each possible MDP. However, this formulation is difficult to solve directly as it requires optimising over the set of variables $\Xi$, which can have infinite dimension since the space of plausible transition functions may be infinite. Thus, we formulate CVaR optimisation in BAMDPs in a way that is more amenable to optimisation: an SG played on an augmented version of the BAMDP. The agent plays against an adversary which modifies the transition probabilities of the original BAMDP. Perturbing the BAMDP transition probabilities is equivalent to perturbing both the prior distribution and each transition function as considered by Prop. 1. This extends work on CVaR optimisation in standard MDPs [8] to the Bayes-adaptive setting.

We begin by augmenting the state space of the BAMDP with an additional state factor, $y \in (0, 1]$, so that now the augmented state comprises $(s, h, y) \in S_{\mathcal{G}}^{+} = S \times \mathcal{H} \times (0, 1]$. The stochastic game proceeds as follows. Given an augmented state, $(s, h, y) \in S_{\mathcal{G}}^{+,agt}$, the agent applies an action, $a \in A$, and receives reward $R_{\mathcal{G}}^{+}((s, h, y), a) = R(s, a)$. The state then transitions to an adversary state $(s, ha, y) \in S_{\mathcal{G}}^{+,adv}$. The adversary then chooses an action from a continuous action space, $\xi \in \Xi(s, a, h, y)$, to perturb the BAMDP transition probabilities where

$$\Xi(s, a, h, y) = \left\{ \xi : S \to \mathbb{R}_{\geq 0} \mid 0 \leq \xi(s') \leq 1/y \ \forall s' \text{ and } \sum_{s' \in S} \xi(s') \cdot T^{+}((s, h), a, (s', has')) = 1 \right\}. \tag{11}$$

In Eq. 11, the adversary perturbation is restricted to adhere to the budget defined by Eq. 3 so that the probability of any history in the BAMDP is perturbed by at most $1/y$. After the adversary chooses the perturbation action, the game transitions back to an agent state $(s', has', y \cdot \xi(s')) \in S_{\mathcal{G}}^{+,agt}$ according to the transition function

$$T_{\mathcal{G}}^{+}((s, ha, y), \xi, (s', has', y \cdot \xi(s'))) = \xi(s') \cdot T^{+}((s, h), a, (s', has')). \tag{12}$$

The initial augmented state of the stochastic game is $s_{0,\mathcal{G}}^{+} = (s_0, h_0, \alpha) \in S_{\mathcal{G}}^{+,agt}$. The tuple $\mathcal{G}^{+} = (S_{\mathcal{G}}^{+}, A_{\mathcal{G}}^{+}, T_{\mathcal{G}}^{+}, R_{\mathcal{G}}^{+}, H, s_{0,\mathcal{G}}^{+})$ defines the Bayes-Adaptive CVaR Stochastic Game (BA-CVaR-SG), where the joint action space is $A_{\mathcal{G}}^{+} = A \cup \Xi$.

The maximin expected value equilibrium of the BA-CVaR-SG optimises the expected value of the BAMDP under perturbations to the probability distribution over paths by an adversary subject to the constraint defined by Eq. 3. Therefore, Prop. 2 states that the maximin expected value equilibrium in the BA-CVaR-SG corresponds to the solution to CVaR optimisation in BAMDPs (Problem 2).

**Proposition 2** *Formulation of CVaR optimisation in BAMDPs as a stochastic game.*

$$\max_{\pi \in \Pi^{\mathcal{M}^{+}}} \text{CVaR}_{\alpha}(G_{\pi}^{\mathcal{M}^{+}}) = \max_{\pi \in \Pi^{\mathcal{G}^{+}}} \min_{\sigma \in \Sigma^{\mathcal{G}^{+}}} \mathbb{E}\left[ G_{(\pi, \sigma)}^{\mathcal{G}^{+}} \right]. \tag{13}$$

Prop. 2 directly extends Prop. 1 from [8] to BAMDPs. The BAMDP transition function is determined by both the prior distribution over transition functions and the transition functions themselves (Eq. 7). Therefore, perturbing the BAMDP transition function, as considered by Prop. 2, is equivalent to perturbing both the prior distribution and each transition function as considered by Prop. 1.

## 4.3 Monte Carlo Tree Search Algorithm

Solving the BA-CVaR-SG is still computationally difficult due to two primary reasons. First, the set of reachable augmented states grows exponentially with the horizon due to the exponentially growing set of possible histories. This prohibits the use of the value iteration approach proposed by Chow et al. [8] for standard MDPs. Second, CVaR is considerably more difficult to optimise than expected value. In the BA-CVaR-SG formulation, this difficulty is manifested in the continuous action space of the adversary corresponding to possible perturbations to the BAMDP transition function.

In this section we present an approximate algorithm, Risk-Averse Bayes-Adaptive Monte Carlo Planning (RA-BAMCP). Like BAMCP [16], which applies MCTS to BAMDPs, and applications of

**Algorithm 1:** Risk-Averse Bayes-Adaptive Monte Carlo Planning

**function** Search($\mathcal{G}^+$)
  create root node $v_0$ with:
    $s_{\mathcal{G}}^+(v_0) = s_{0,\mathcal{G}}^+$
    $player(v_0) = agent$
  **while** within computational budget **:**
    $v_{leaf} \leftarrow \texttt{TreePolicy}(v_0)$
    $\widetilde{r} \leftarrow \texttt{DefaultPolicy}(v_{leaf})$
    $\texttt{UpdateNodes}(v_{leaf}, \widetilde{r})$
  $v' = \underset{v' \in \text{children of } v_0}{\arg\max} \widehat{Q}(v')$
  $v'' = \underset{v'' \in \text{children of } v'}{\arg\min} \widehat{Q}(v'')$
  **return** $a(v'), \xi(v'')$

**function** TreePolicy($v$)
  **while** $v$ is non-terminal **:**
    **if** $player(v) = agent$ **:**
      **if** $v$ not fully expanded **:**
        **return** ExpandAgent($v$)
      **else**
        $v \leftarrow v.\texttt{BestChild}()$
    **else if** $player(v) = adv$ **:**
      **if** $(N(v))^\tau \geq |\widetilde{\Xi}(v)|$ **:**
        **return** ExpandAdversary($v$)
      **else**
        $v \leftarrow v.\texttt{BestChild}()$
    **else if** $player(v) = chance$ **:**
      $v \leftarrow$ sample successor according to Eq.
      12

**function** UpdateNodes($v, \widetilde{r}$)
  **while** $v$ is not null **:**
    $N(v) \mathrel{+}= 1$
    $\widehat{Q}(v) \mathrel{+}= \frac{\widetilde{r}(v) - \widehat{Q}(v)}{N(v)}$
    $v \leftarrow$ parent of $v$

**function** ExpandAgent($v$)
  choose $a \in$ untried actions from action set $A$
  add child $v'$ to $v$ with:
    $s_{\mathcal{G}}^+(v') = s_{\mathcal{G}}^+(v)$
    $a(v') = a$
    $player(v') = adv$
    $\widetilde{\Xi}(v') = \varnothing$
  **return** $v'$

**function** ExpandAdversary($v$)
  **if** $\widetilde{\Xi}(v) = \varnothing$ **:**
    $\xi_{new} = \texttt{RandomPerturbation}(v)$
  **else**
    $\xi_{new} = \texttt{BayesOptPerturbation}(v)$
  $\widetilde{\Xi}(v).\texttt{insert}(\xi_{new})$
  add child $v'$ to $v$ with:
    $s_{\mathcal{G}}^+(v') = s_{\mathcal{G}}^+(v)$
    $\xi(v') = \xi$
    $player(v') = chance$
  **return** $v'$

**function** BestChild($v$)
  **if** $player(v) = agent$ **:**
    **return**
      $\underset{v' \in \text{children of } v}{\arg\max} \left[ \widehat{Q}(v') + c_{mcts} \cdot \sqrt{\frac{\log N(v)}{N(v')}} \right]$
  **else**
    **return**
      $\underset{v' \in \text{children of } v}{\arg\min} \left[ \widehat{Q}(v') - c_{mcts} \cdot \sqrt{\frac{\log N(v)}{N(v')}} \right]$

MCTS to two-player games such as Go [14], we use MCTS to handle the large state space by focusing search in promising areas. Unlike BAMCP, we perform search over a Bayes-adaptive stochastic game against an adversary which may perturb the probabilities, and we cannot use root sampling as we need access to the BAMDP transition dynamics (Eq. 7) to compute the set of admissible adversary actions (Eq. 11). To handle the continuous action space for the adversary, we use progressive widening [10] paired with Bayesian optimisation [25, 22] to prioritise promising actions.

**RA-BAMCP Outline** Algorithm 1 proceeds using the standard components of MCTS: selecting nodes within the tree, expanding leaf nodes, performing rollouts using a default policy, and updating the statistics in the tree. The search tree consists of alternating layers of three types of nodes: agent decision nodes, adversary decision nodes, and chance nodes. At each new node $v$, the value estimate associated with that node, $\widehat{Q}(v)$, and visitation count, $N(v)$, are both initialised to zero. At agent (adversary) decision nodes we use an upper (lower) confidence bound to optimistically select actions within the tree. At newly expanded chance nodes we compute $\mathcal{P}(T|h)$, the posterior distribution over the transition function for the history leading up to that chance node. Using this posterior distribution, we can compute the transition function for the BA-CVaR-SG, $T_{\mathcal{G}}^+$, at that chance node. When a chance node is reached during a simulation, the successor state is sampled using $T_{\mathcal{G}}^+$.

At each adversary node, a discrete set of perturbation actions is available to the adversary, denoted by $\widetilde{\Xi}(v)$. We use progressive widening [10] to gradually increase the set of perturbation actions available by adding actions from the continuous set of possible actions, $\Xi(v)$. The rate at which new actions are expanded is controlled by the progressive widening parameter, $\tau$. A naive implementation of

progressive widening would simply sample new perturbation actions randomly. This might require many expansions before finding a good action. To mitigate this issue we use Gaussian process (GP) Bayesian optimisation to select promising actions to expand in the tree [25, 22].

In our algorithm the `BayesOptPerturbation` function performs Bayesian optimisation to return a promising perturbation action at a given adversary node, $v$. We train a GP using a Gaussian kernel. The input features are existing expanded perturbation actions at the node, $\xi \in \widetilde{\Xi}(v)$. The number of input dimensions is equal to the number of successor states with non-zero probability according to the BAMDP transition function. The training labels are the corresponding value estimates in the tree for each of the perturbation actions: $\widehat{Q}(v)$, for all children $v'$ of $v$. The acquisition function for choosing the new perturbation action to expand is a lower confidence bound which is optimistic for the minimising adversary: $\xi_{new} = \arg\min_\xi [\mu(\xi) - c_{bo} \cdot \sigma(\xi)]$, where $\mu(\xi)$ and $\sigma(\xi)$ are the mean and standard deviations of the GP predicted posterior distribution respectively. Prop. 3 establishes that a near-optimal perturbation will eventually be expanded with high probability if $c_{bo}$ is set appropriately.

**Proposition 3** *Let $\delta \in (0, 1)$. Provided that $c_{bo}$ is chosen appropriately (details in the appendix), as the number of perturbations expanded approaches $\infty$, a perturbation within any $\epsilon > 0$ of the optimal perturbation will be expanded by the Bayesian optimisation procedure with probability $\geq 1 - \delta$.*

After each simulation, the value estimates at each node are updated using $\widetilde{r}(v)$, the reward received from $v$ onwards during the simulation. Search continues by performing simulations until the computational budget is exhausted. At this point, the estimated best action for the agent and the adversary is returned. In our experiments we perform online planning: after executing an action and transitioning to a new state, we perform more simulations from the new root state.

## 5   Experiments

Code for the experiments is included in the supplementary material. All algorithms are implemented in C++ and Gurobi is used to solve linear programs for the value iteration (VI) approaches. Computation times are reported for a 3.2 GHz Intel i7 processor with 64 GB of RAM.

No existing works address the problem formulation in this paper. Therefore, we adapt two methods for CVaR optimisation to the BAMDP setting and also compare against approaches for related problems to assess their performance in our setting. We compare the following approaches:

- *RA-BAMCP*: the approach presented in this paper.

- *CVaR BAMDP Policy Gradient* (PG): applies the policy gradient approach to CVaR optimisation from [45] with the vectorised belief representation for BAMDPs proposed in [15] with linear function approximation. Further details on this approach can be found in the supplementary material.

- *CVaR VI BAMDP*: the approximate VI CVaR optimisation approach from [8] applied to the BAMDP. Due to the extremely poor scalability of this approach we only test it on the smaller of our domains.

- *CVaR VI Expected MDP* (EMDP): the approximate VI approach from [8] applied to the expected MDP parameters according to the prior distribution over transition functions, $\mathcal{P}(T|h_0)$.

- *BAMCP*: Bayes-Adaptive Monte Carlo Planning which optimises the expected value [16]. This is equivalent to *RA-BAMCP* with the confidence level, $\alpha$, set to 1.

For *RA-BAMCP* and *BAMCP* the MCTS exploration parameter was set to $c_{mcts} = 2$ based on empirical performance. For these methods, we performed $100,000$ simulations for the initial step and $25,000$ simulations per step thereafter. For both of these algorithms the rollout policy used for the agent was the *CVaR VI Expected MDP* policy. For *RA-BAMCP*, the rollout policy for the adversary was random, the progressive widening parameter was set to $\tau = 0.2$, and the exploration parameter for the GP Bayesian optimisation was also set to $c_{bo} = 2$. Details of the GP hyperparameters are in the supplementary material. For *CVaR VI Expected MDP* and *CVaR VI BAMDP*, 20 log-spaced points were used to discretise the continuous state space for approximate VI as described by [8]. For *CVaR BAMDP Policy Gradient* we trained using $2 \times 10^6$ simulations and details on hyperparameter tuning are in the supplementary material.

**Betting Game Domain**   We adapt this domain from the literature on conditional value at risk in Markov decision processes [3] to the Bayes-adaptive setting. The agent begins with $money = 10$. At each stage the agent may choose to place a bet from $bets = \{0, 1, 2, 5, 10\}$ provided that sufficient money is available. If the agent wins the game at that stage, an amount of money equal to the bet placed is received. If the agent loses the stage, the amount of money bet is lost. The reward is the total money the agent has after 6 stages of betting. In our Bayes-adaptive setting, the probability of winning the game is not known apriori. Instead, a prior distribution over the probability of winning the game is modelled using a beta distribution with parameters $\alpha = \frac{10}{11}, \beta = \frac{1}{11}$, indicating that the odds for the game are likely to be in favour of the agent.

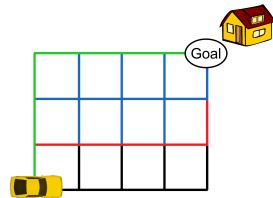

Figure 1: Autonomous car navigation domain. Colours indicate the road types as follows - black: *highway*, red: *main road*, blue: *street*, green: *lane*.

**Autonomous Car Navigation Domain**   An autonomous car must navigate along roads to a destination by choosing from actions {*up, down, left, right*} (Figure 1). There are four types of roads: {*highway, main road, street, lane*} with different transition duration characteristics. The transition distribution for each type of road is unknown. Instead, the prior belief over the transition duration distribution for each road type is modelled by a Dirichlet distribution. Thus, the belief is represented by four Dirichlet distributions. Details on the Dirichlet prior used for each road type are included in the supplementary material. Upon traversing each section of road, the agent receives reward $-time$ where $time$ is the transition duration for traversing that section of road. Upon reaching the destination, the agent receives a reward of 80. The search horizon was set to $H = 10$.

### 5.1   Results

We evaluated the return from each method over 2000 episodes. For each episode, the true underlying MDP was sampled from the prior distribution. Therefore, the return received for each evaluation episode is subject to both epistemic uncertainty due to the uncertain MDP model, and aleatoric uncertainty due to stochastic transitions. Results for this evaluation can be found in Table 1. The rows indicate the method and the confidence level, $\alpha$, that the method is set to optimise. The columns indicate the performance of each method for CVaR with $\alpha = 0.03$ and $\alpha = 0.2$, and expected value ($\alpha = 1$), evaluated using the sample of 2000 episodes. Note that when evaluating the performance of each algorithm, we only care about the CVaR performance for the confidence level that the algorithm is set to optimise.

In the Betting Game domain, strong performance is attained by *CVaR VI BAMDP*, however the computation time required for this approach is considerable. It was not possible to test *CVaR VI BAMDP* on the Autonomous Navigation due to its poor scalability. *RA-BAMCP* attained near-optimal performance at both confidence levels using substantially less computation. *CVaR VI Expected MDP* performed well at the $\alpha = 0.2$ confidence level, but very poorly at $\alpha = 0.03$. This approach performs poorly at $\alpha = 0.03$ because by using the expected value of the transition function, the probability of repeatedly losing the game is underestimated. *BAMCP* performed the best at optimising expected value, but less well than other approaches for optimising CVaR.

*CVaR BAMDP Policy Gradient* performed strongly in the betting domain with $\alpha = 0.2$. However, it attained poor asymptotic performance with $\alpha = 0.03$. Conversely, this method performed well in the navigation domain with $\alpha = 0.03$, but poorly for $\alpha = 0.2$. These results indicate that the policy gradient approach is susceptible to local minima. Training curves for *CVaR BAMDP Policy Gradient* can be found in the supplementary material.

In the Autonomous Car Navigation domain *RA-BAMCP* outperformed *CVaR VI Expected MDP* at both confidence levels. As expected, *BAMCP* again performed the best at optimising expected value, but not CVaR. The return histograms in Figure 2 show that while *BAMCP* optimises expected value, there is high uncertainty over the return value. As the confidence level $\alpha$ is decreased, applying *RA-BAMCP* decreases the likelihood of poor returns by shifting the left tail of the return distribution to the right, at the expense of worse expected performance. This demonstrates that *RA-BAMCP* is risk-averse to poor return values in the presence of both epistemic and aleatoric uncertainty.

| Method | Time (s) | CVaR ($\alpha=0.03$) | CVaR ($\alpha=0.2$) | Expected Value |
|---|---|---|---|---|
| *RA-BAMCP* ($\alpha=0.03$) | 17.9 (0.2) | 9.98 (0.02) | 10.00 (0.002) | 10.00 (0.001) |
| *CVaR VI BAMDP* ($\alpha=0.03$) | 8620.5 | **10.00** (0.0) | 10.00 (0.0) | 10.00 (0.0) |
| *CVaR VI EMDP* ($\alpha=0.03$) | 61.7 | 0.00 (0.0) | 10.45 (0.33) | 15.72 (0.09) |
| *CVaR BAMDP PG* ($\alpha=0.03$) | 4639.3 | 2.90 (0.06) | 10.57 (0.33) | 27.34 (0.24) |
| *RA-BAMCP* ($\alpha=0.2$) | 78.9 (0.2) | 0.00 (0.0) | 20.09 (1.04) | 46.84 (0.37) |
| *CVaR VI BAMDP* ($\alpha=0.2$) | 8620.5 | 0.00 (0.0) | **20.77** (1.02) | 44.15 (0.33) |
| *CVaR VI EMDP* ($\alpha=0.2$) | 61.7 | 0.00 (0.0) | 19.33 (0.99) | 39.86 (0.30) |
| *CVaR BAMDP PG* ($\alpha=0.2$) | 4309.7 | 0.00 (0.0) | 19.85 (0.97) | 46.65 (0.37) |
| *BAMCP* | 13.5 (0.01) | 0.00 (0.0) | 17.07 (1.09) | **59.36** (0.52) |

(a) Betting Game domain

| Method | Time (s) | CVaR ($\alpha=0.03$) | CVaR ($\alpha=0.2$) | Expected Value |
|---|---|---|---|---|
| *RA-BAMCP* ($\alpha=0.03$) | 255.7 (0.2) | 23.51 (0.17) | 25.41 (0.06) | 29.56 (0.06) |
| *CVaR VI EMDP* ($\alpha=0.03$) | 96.6 | 20.65 (0.68) | 26.11 (0.16) | 30.13 (0.07) |
| *CVaR BAMDP PG* ($\alpha=0.03$) | 37232.6 | **23.92** (0.08) | 25.38 (0.05) | 28.97 (0.05) |
| *RA-BAMCP* ($\alpha=0.2$) | 205.1 (0.3) | 18.91 (0.44) | **27.76** (0.24) | 38.33 (0.15) |
| *CVaR VI EMDP* ($\alpha=0.2$) | 96.6 | 4.20 (0.54) | 20.42 (0.44) | 36.64 (0.21) |
| *CVaR BAMDP PG* ($\alpha=0.2$) | 36686.1 | 10.05 (0.74) | 24.70 (0.39) | 49.67 (0.36) |
| *BAMCP* | 74.4 (0.02) | 5.56 (0.60) | 21.46 (0.47) | **50.16** (0.38) |

(b) Autonomous Car Navigation domain

Table 1: Results from evaluating the returns of each method for 2000 episodes. Brackets indicate standard errors. Time for *RA-BAMCP* and *BAMCP* indicates average total computation time per episode to perform simulations. Time for *CVaR VI BAMDP* and *CVaR VI Expected MDP* indicates the time for VI to converge. Time for *CVaR BAMDP PG* indicates the total training time.

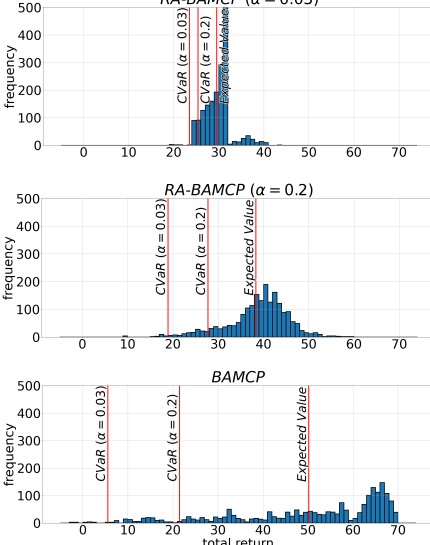

Figure 2: Histogram of returns received in the Autonomous Car Navigation domain. Vertical red lines indicate $CVaR_{0.03}$, $CVaR_{0.2}$, and the expected value of the return distributions.

Results in the supplementary material show that our approach using random action expansion rather than Bayesian optimisation attains worse performance for the same computation budget. This demonstrates that Bayesian optimisation for action expansion is critical to our approach.

## 6 Discussion of Limitations and Conclusion

In this work, we have addressed CVaR optimisation in the BAMDP formulation of model-based Bayesian RL. This is motivated by the aim of mitigating risk due to both epistemic uncertainty over the model, and aleatoric uncertainty due to environment stochasticity. Our experimental evaluation demonstrates that our algorithm, *RA-BAMCP*, outperforms baseline approaches for this problem.

One of the limitations of our work is that our approach cannot use root sampling [16]. Instead, the posterior distribution is computed at each node in the MCTS tree, prohibiting the use of complex priors. In future work, we wish to develop an MCTS algorithm for CVaR optimisation which can use root sampling. Additionally, due to the exponential growth of the BAMDP and the difficulty of optimising CVaR, the scalability of our approach is limited. In future work, we wish to utilise function approximation to improve the scalability of our approach to more complex problems. One direction could be using a neural network to guide search, in the style of AlphaGo [39]. Another direction could be to extend the policy gradient algorithm that we proposed in the experiments section to make use of deep learning.

## Acknowledgements and Funding Disclosure

The authors would like to thank Paul Duckworth for valuable feedback on an earlier draft of this work.

This work was supported by a Programme Grant from the Engineering and Physical Sciences Research Council [EP/V000748/1], the Clarendon Fund at the University of Oxford, and a gift from Amazon Web Services.

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
