# A  Proof of Proposition 1

Let $\mathcal{M}^+$ be a BAMDP. Then:

$$\text{CVaR}_\alpha(G_\pi^{\mathcal{M}^+}) = \min_{\substack{\delta:\mathcal{T}\to\mathbb{R}_{\geq 0} \\ \xi\in\Xi}} \mathbb{E}\big[G_\pi^{\mathcal{M}^+_{\delta,\xi}}\big], \quad \textbf{s.t.} \tag{14}$$

$$0 \leq \delta(T_p)\xi_p(s_0,a_0,s_1)\xi_p(s_1,a_1,s_2)\cdots\xi_p(s_{H\text{-}1},a_{H\text{-}1},s_H) \leq \frac{1}{\alpha},$$

$$\forall\, T_p \in \mathcal{T},\ \forall(s_0,a_0,s_1,a_1,\ldots,s_H)\in\mathcal{H}_H.$$

*Proof*: We denote by $\mathcal{P}_\pi^{\mathcal{M}^+}(h_H)$ the probability of history $h_H$ in BAMDP $\mathcal{M}^+$, under policy $\pi$. This probability can be expressed as

$$\mathcal{P}_\pi^{\mathcal{M}^+}(h_H) = T^+\big((s_0,h_0),\pi(h_0),(s_1,h_1)\big)T^+\big((s_1,h_1),\pi(h_1),(s_2,h_2)\big)\ldots T^+\big((s_{H\text{-}1},h_{H\text{-}1}),\pi(h_{H\text{-}1}),(s_H,h_H)\big). \tag{15}$$

By substituting in the transition function for the BAMDP from Equation 7 we have

$$\mathcal{P}_\pi^{\mathcal{M}^+}(h_H) = \int_T T(s_0,\pi(h_0),s_1)\mathcal{P}(T|h_0)dT \cdot \int_T T(s_1,\pi(h_1),s_2)\mathcal{P}(T|h_1)dT\ldots$$
$$\int_T T(s_{H\text{-}1},\pi(h_{H\text{-}1}),s_H)\mathcal{P}(T|h_{H-1})dT. \tag{16}$$

Bayes' rule states that

$$\mathcal{P}(T|h_1) = \frac{T(s_0,\pi(h_0),s_1)\cdot\mathcal{P}(T|h_0)}{\int_{T'} T'(s_0,\pi(h_0),s_1)\cdot\mathcal{P}(T'|h_0)dT'}. \tag{17}$$

Substituting Bayes' posterior update into Equation 16 we have

$$\mathcal{P}_\pi^{\mathcal{M}^+}(h_H) = \int_T T(s_0,\pi(h_0),s_1)\mathcal{P}(T|h_0)dT\cdot\int_T T(s_1,\pi(h_1),s_2)\frac{T(s_0,\pi(h_0),s_1)\cdot\mathcal{P}(T|h_0)}{\int_{T'} T'(s_0,\pi(h_0),s_1)\cdot\mathcal{P}(T'|h_0)dT'}dT\ldots$$
$$\int_T T(s_{H\text{-}1},\pi(h_{H\text{-}1}),s_H)\mathcal{P}(T|h_{H-1})dT = \int_T T(s_0,\pi(h_0),s_1)\cdot T(s_1,\pi(h_1),s_2)\cdot\mathcal{P}(T|h_0)dT\ldots$$
$$\int_T T(s_{H\text{-}1},\pi(h_{H\text{-}1}),s_H)\mathcal{P}(T|h_{H-1})dT. \tag{18}$$

Repeating this process of substitution and cancellation we get

$$\mathcal{P}_\pi^{\mathcal{M}^+}(h_H) = \left[\int_T T(s_0,\pi(h_0),s_1)\cdot T(s_1,\pi(h_1),s_2)\ldots T(s_{H\text{-}1},\pi(h_{H\text{-}1}),s_H)\cdot\mathcal{P}(T|h_0)dT\right]. \tag{19}$$

Define $\mathcal{T}$ to be the support of $\mathcal{P}(T|h_0)$, and let $T_p \in \mathcal{T}$ be a plausible transition function. Now consider a perturbation of the prior distribution over transition functions $\delta : \mathcal{T} \to \mathbb{R}_{\geq 0}$ such that $\int_{T_p} \delta(T_p)\mathcal{P}(T_p|h_0)dT_p = 1$. Additionally, for $T_p \in \mathcal{T}$, consider a perturbation of $T_p$, $\xi_p :$ $S \times A \times S \to \mathbb{R}_{\geq 0}$ such that $\sum_{s'\in S} \xi_p(s,a,s')\cdot T_p(s,a,s') = 1\ \forall s,a$. We denote the set of all possible perturbations of $T_p$ as $\Xi_p$, and the set of all perturbations over $\mathcal{T}$ as $\Xi = \times_{T_p\in\mathcal{T}}\Xi_p$. For BAMDP $\mathcal{M}^+$, $\delta : \mathcal{T} \to \mathbb{R}_{\geq 0}$, and $\xi \in \Xi$, we define $\mathcal{M}^+_{\delta,\xi}$ as the BAMDP, obtained from $\mathcal{M}^+$, with perturbed prior distribution $\mathcal{P}^\delta(T_p|h_0) = \delta(T_p)\cdot\mathcal{P}(T_p|h_0)$ and perturbed transition functions

$T_p^\xi(s, a, s') = \xi_p(s, a, s') \cdot T_p(s, a, s')$. We denote by $\mathcal{P}_\pi^{\mathcal{M}_{\delta,\xi}^+}(h_H)$ the probability of $h_H$ in the perturbed BAMDP, $\mathcal{M}_{\delta,\xi}^+$, which can be expressed as follows

$$
\begin{aligned}
\mathcal{P}_\pi^{\mathcal{M}_{\delta,\xi}^+}(h_H) &= \left[ \int_{T_p} T_p^\xi(s_0, \pi(h_0), s_1) \cdot T_p^\xi(s_1, \pi(h_1), s_2)) \dots T_p^\xi(s_{H\text{-}1}, \pi(h_{H\text{-}1}), s_H) \cdot \mathcal{P}^\delta(T_p|h_0) dT_p \right] \\
&= \left[ \int_{T_p} \xi_p(s_0, \pi(h_0), s_1) \cdot T_p(s_0, \pi(h_0), s_1) \cdot \xi_p(s_1, \pi(h_1), s_2) \cdot T_p(s_1, \pi(h_1), s_2)) \dots \right. \\
&\qquad \left. \cdot \xi_p(s_{H\text{-}1}, \pi(h_{H\text{-}1}), s_H) \cdot T_p(s_{H\text{-}1}, \pi(h_{H\text{-}1}), s_H) \cdot \delta(T_p) \cdot \mathcal{P}(T_p|h_0) dT_p \right] \\
&= \left[ \int_{T_p} \xi_p(s_0, \pi(h_0), s_1) \cdot \xi_p(s_1, \pi(h_1), s_2) \dots \xi_p(s_{H\text{-}1}, \pi(h_{H\text{-}1}), s_H) \cdot \delta(T_p) \cdot \right. \\
&\qquad \left. T_p(s_0, \pi(h_0), s_1) \cdot T_p(s_1, \pi(h_1), s_2)) \dots T_p(s_{H\text{-}1}, \pi(h_{H\text{-}1}), s_H) \cdot \mathcal{P}(T_p|h_0) dT_p \right] \quad (20)
\end{aligned}
$$

Let $\kappa_{\delta,\xi}(h_H) = \dfrac{\mathcal{P}_\pi^{\mathcal{M}_{\delta,\xi}^+}(h_H)}{\mathcal{P}_\pi^{\mathcal{M}^+}(h_H)}$ denote the total perturbation to the probability of any path. Provided that the perturbations satisfy the condition

$$
0 \le \delta(T_p)\xi_p(s_0, a_0, s_1)\xi_p(s_1, a_1, s_2) \cdots \xi_p(s_{H\text{-}1}, a_{H\text{-}1}, s_H) \le \frac{1}{\alpha},
$$

$$
\forall\, T_p \in \mathcal{T},\ \forall(s_0, a_0, s_1, a_1, \dots, s_H) \in \mathcal{H}_H,
$$

we observe from Equation 20 that the perturbation to the probability of any path is bounded by

$$
0 \le \kappa_{\delta,\xi}(h_H) \le \frac{1}{\alpha}. \quad (21)
$$

The expected value of the perturbation to the probability of any path is

$$
\begin{aligned}
\mathbb{E}[\kappa_{\delta,\xi}(h_H)] &= \sum_{h_H \in \mathcal{H}_H} \left[ \mathcal{P}_\pi^{\mathcal{M}^+}(h_H) \cdot \kappa_{\delta,\xi}(h_H) \right] = \sum_{h_H \in \mathcal{H}_H} \left[ \mathcal{P}_\pi^{\mathcal{M}^+}(h_H) \frac{\mathcal{P}_\pi^{\mathcal{M}_{\delta,\xi}^+}(h_H)}{\mathcal{P}_\pi^{\mathcal{M}^+}(h_H)} \right] \\
&= \sum_{h_H \in \mathcal{H}_H} \left[ \mathcal{P}_\pi^{\mathcal{M}_{\delta,\xi}^+}(h_H) \right] = 1, \quad (22)
\end{aligned}
$$

where the last equality holds because the conditions $\sum_{s' \in S} \xi_p(s, a, s') \cdot T_p(s, a, s') = 1\ \forall s, a, p$ and $\int_{T_p} \delta(T_p)\mathcal{P}(T_p|h_0) dT_p = 1$ ensure that the distribution over paths in the perturbed BAMDP, $\mathcal{M}_{\delta,\xi}^+$, is a valid probability distribution. Therefore,

$$
\begin{aligned}
\min_{\substack{\delta:\mathcal{T}\to\mathbb{R}_{\ge 0} \\ \xi\in\Xi}} \mathbb{E}\big[G_\pi^{\mathcal{M}_{\delta,\xi}^+}\big] &= \min_{\substack{\delta:\mathcal{T}\to\mathbb{R}_{\ge 0} \\ \xi\in\Xi}} \sum_{h_H} \left[ \mathcal{P}_\pi^{\mathcal{M}_{\delta,\xi}^+}(h_H) \cdot \widetilde{r}(h_H) \right] = \\
&\min_{\substack{\kappa_{\delta,\xi}, \ \text{s.t.} \\ 0\le\kappa_{\delta,\xi}(h_H)\le\frac{1}{\alpha} \\ \mathbb{E}[\kappa_{\delta,\xi}(h_H)]=1}} \sum_{h_H} \left[ \mathcal{P}_\pi^{\mathcal{M}^+}(h_H) \cdot \kappa_{\delta,\xi}(h_H) \cdot \widetilde{r}(h_H) \right] = \mathrm{CVaR}_\alpha(G_\pi^{\mathcal{M}^+}) \quad (23)
\end{aligned}
$$

where $\widetilde{r}(h_H)$ indicates the total sum of rewards for history $h_H$, and the last equality holds from the CVaR dual representation theorem stated in Equations 2 and 3 [34, 37]. $\square$

# B Proof of Proposition 2

Formulation of CVaR optimisation in BAMDPs as a stochastic game.

$$\max_{\pi \in \Pi^{\mathcal{M}^+}} \text{CVaR}_\alpha(G_\pi^{\mathcal{M}^+}) = \max_{\pi \in \Pi^{\mathcal{G}^+}} \min_{\sigma \in \Sigma^{\mathcal{G}^+}} \mathbb{E}\big[G_{(\pi,\sigma)}^{\mathcal{G}^+}\big]. \tag{24}$$

*Proof*: Proposition 2 directly extends Proposition 1 in [8] to BAMDPs. Let $\mathcal{P}_\pi^{\mathcal{M}^+}(h_H)$ denote the probability of history $h_H$ in BAMDP $\mathcal{M}^+$ under policy $\pi$ as defined in Equation 15. We denote by $\mathcal{P}_{\pi,\sigma}^{\mathcal{G}^+}(h_H)$ the probability of history $h_H$ by following policy $\pi$ and adversary perturbation policy $\sigma$ in BA-CVaR-SG, $\mathcal{G}^+$. This probability can be expressed as

$$\mathcal{P}_{\pi,\sigma}^{\mathcal{G}^+}(h_H) = T^+\big((s_0,h_0), \pi(h_0), (s_1,h_1)\big) \cdot \xi\big((s_0,h_0), \pi(h_0), s_1\big) \cdot T^+\big((s_1,h_1), \pi(h_1), (s_2,h_2)\big) \cdot \xi\big((s_1,h_1), \pi(h_1), s_2\big)$$
$$\dots T^+\big((s_{H\text{-}1}, h_{H\text{-}1}), \pi(h_{H\text{-}1}), (s_H, h_H)\big) \cdot \xi\big((s_{H\text{-}1}, h_{H\text{-}1}), \pi(h_{H\text{-}1}), s_H\big), \tag{25}$$

where $\xi\big((s,h), a, s'\big)$ indicates the perturbation applied by $\sigma$ for successor state $s'$ after action $a$ is executed in augmented state $(s,h)$.

Let $\kappa_\sigma(h_H)$ denote the total perturbation by the adversary to the probability of any history in the BAMDP,

$$\kappa_\sigma(h_H) = \frac{\mathcal{P}_{\pi,\sigma}^{\mathcal{G}^+}(h_H)}{\mathcal{P}_\pi^{\mathcal{M}^+}(h_H)} = \xi\big((s_0,h_0), \pi(h_0), s_1\big) \cdot \xi\big((s_1,h_1), \pi(h_1), s_2\big) \dots \xi\big((s_{H\text{-}1}, h_{H\text{-}1}), \pi(h_{H\text{-}1}), s_H\big). \tag{26}$$

By the definition of the admissible adversary perturbations in Equation 11 the perturbation to the probability of any path is bounded by

$$0 \leq \kappa_\sigma(h_H) \leq \frac{1}{\alpha}. \tag{27}$$

The expected value of the perturbation to the probability of any path is

$$\mathbb{E}[\kappa_\sigma(h_H)] = \sum_{h_H \in \mathcal{H}_H} \Big[ \mathcal{P}_\pi^{\mathcal{M}^+}(h_H) \cdot \kappa_\sigma(h_H) \Big] = \sum_{h_H \in \mathcal{H}_H} \Big[ \mathcal{P}_\pi^{\mathcal{M}^+}(h_H) \frac{\mathcal{P}_{\pi,\sigma}^{\mathcal{G}^+}(h_H)}{\mathcal{P}_\pi^{\mathcal{M}^+}(h_H)} \Big]$$
$$= \sum_{h_H \in \mathcal{H}_H} \Big[ \mathcal{P}_{\pi,\sigma}^{\mathcal{G}^+}(h_H) \Big] = 1, \tag{28}$$

where the last equality holds because Equation 11 ensures that the perturbed transition probabilities are a valid probability distribution. Therefore, the perturbed distribution over histories is also a valid probability distribution. Therefore,

$$\max_{\pi \in \Pi^{\mathcal{G}^+}} \min_{\sigma \in \Sigma^{\mathcal{G}^+}} \mathbb{E}\big[G_{(\pi,\sigma)}^{\mathcal{G}^+}\big] = \max_{\pi \in \Pi^{\mathcal{G}^+}} \min_{\sigma \in \Sigma^{\mathcal{G}^+}} \sum_{h_H \in \mathcal{H}_H} \Big[ \mathcal{P}_{\pi,\sigma}^{\mathcal{G}^+}(h_H) \cdot \widetilde{r}(h_H) \Big] =$$
$$\max_{\pi \in \Pi^{\mathcal{G}^+}} \min_{\substack{\kappa_\sigma, \text{ s.t.} \\ 0 \leq \kappa_\sigma(h_H) \leq \frac{1}{\alpha} \\ \mathbb{E}[\kappa_\sigma(h_H)]=1}} \sum_{h_H} \Big[ \mathcal{P}_\pi^{\mathcal{M}^+}(h_H) \cdot \kappa_\sigma(h_H) \cdot \widetilde{r}(h_H) \Big] = \max_{\pi \in \Pi^{\mathcal{G}^+}} \text{CVaR}_\alpha(G_\pi^{\mathcal{M}^+}). \tag{29}$$

where $\widetilde{r}(h_H)$ indicates the total sum of rewards for history $h_H$, and the last equality holds from the CVaR dual representation theorem stated in Equations 2 and 3 [34, 37]. $\square$

# C Proof of Proposition 3

Let $\delta \in (0,1)$. Provided that $c_{bo}$ is chosen appropriately (details in the appendix), as the number of perturbations expanded approaches $\infty$, a perturbation within any $\epsilon > 0$ of the optimal perturbation will be expanded by the Bayesian optimisation procedure with probability $\geq 1 - \delta$.

*Proof*: Consider an adversary decision node, $v$, associated with augmented state $(s, ha, y)$ in the BA-CVaR-SG. Let $Q((s, ha, y), \xi)$ denote the true minimax optimal value of applying adversary perturbation $\xi$ at augmented state $(s, ha, y)$ in the BA-CVaR-SG. We begin by proving that $Q((s, ha, y), \xi)$ is continuous with respect to $\xi$. Define a function $d : S \to \mathbb{R}$, such that $\xi + d$ produces a valid adversary perturbation. We can write $Q((s, ha, y), \xi + d)$ as a sum over successor states

$$Q((s, ha, y), \xi + d) = \sum_{s'} (\xi(s') + d(s')) \cdot T^+((s, h), a, (s', has')) \cdot V(s', has', y \cdot (\xi(s') + d(s'))), \tag{30}$$

where $V(s, h, y)$ denotes the optimal expected value in the BA-CVaR-SG at augmented state $(s, h, y)$. Because $V(s, h, y)$ equals the CVaR at confidence level $y$ when starting from $(s, h)$, and CVaR is continuous with respect to the confidence level [34], we have that

$$\lim_{(d(s_1), \ldots, d(s_{|S|})) \to (0, \ldots, 0)} Q((s, ha, y), \xi + d) = \sum_{s'} \xi(s') \cdot T^+((s, h), a, (s', has')) \cdot V(s', has', y \cdot \xi(s'))$$
$$= Q((s, ha, y), \xi). \tag{31}$$

Therefore, $Q((s, ha, y), \xi)$ is continuous with respect to $\xi$. Now, consider the Gaussian kernel which we use for Bayesian optimisation, $k : \Xi \times \Xi \to \mathbb{R}$, where $\Xi$ is the compact set of admissible adversary actions according to Equation 11

$$k(\xi, \xi') = \exp \left( - \frac{||\xi - \xi'||^2}{2l^2} \right). \tag{32}$$

Because the Gaussian kernel is a "universal" kernel [23], the reproducing kernel Hilbert space (RKHS), $\mathcal{H}_k$, corresponding to kernel $k$ is dense in $\mathcal{C}(\Xi)$, the set of real-valued continuous functions on $\Xi$. Because $Q((s, ha, y), \xi)$ is continuous with $\xi$, $Q((s, ha, y), \xi)$ belongs to the RKHS of $k$. Because the true underlying function that we are optimising belongs to the RKHS of the kernel, Theorem 3 from [41] applies.

Consider minimising the true underlying function $Q((s, ha, y), \xi)$ by using GP Bayesian optimisation to decide the next adversary perturbation $\xi \in \Xi$ to sample. Our prior is $GP(0, k(\xi, \xi'))$, and noise model $N(0, \sigma_n^2)$. For each perturbation expanded so far at $v$, $\xi \in \widetilde{\Xi}(v)$, we assume that the $Q$ value of $\xi$ estimated using MCTS, $\widehat{Q}((s, ha, y), \xi)$, is a noisy estimate of the true optimal $Q$ value, i.e. $\widehat{Q}((s, ha, y), \xi) = Q((s, ha, y), \xi) + \epsilon_n$, where the noise is bounded by $\epsilon_n < \sigma_n$. For each round, $t = 1, 2, \ldots$, we choose the new perturbation to be expanded using a lower confidence bound, $\xi_t = \arg\min_\xi [\mu_{t-1}(\xi) - c_{bo}\sigma_{t-1}(\xi)]$, where $\mu_{t-1}(\xi)$, and $\sigma_{t-1}(\xi)$ are the mean and standard deviation of the GP posterior distribution after performing Bayesian updates using the $Q$-value estimates, $\widehat{Q}((s, ha, y), \xi)$, up to $t - 1$.

We define the cumulative regret after $T$ rounds of expanding new perturbation actions as

$$R_T = \sum_{t=1}^{T} Q((s, ha, y), \xi_t) - Q((s, ha, y), \xi^*), \tag{33}$$

where $\xi^*$ is the optimal adversary perturbation at $(s, ha, y)$. Theorem 3 from [41] states the following. Let $\delta \in (0, 1)$. Assume $||Q((s, ha, y), \cdot)||_k^2 \leq B$, where $||f||_k$ denotes the RKHS norm of $f$ induced by $k$. Set the exploration parameter in the lower confidence bound acquisition function

to $c_{bo} = \sqrt{2B + 300\gamma_t \log(t/\delta)^3}$, where $\gamma_t$ is the maximum information gain at round $t$, which is defined and bounded in [41]. Then the following holds

$$\Pr\left\{R_T \leq \sqrt{C_1 T \beta_T \gamma_T} \quad \forall T \geq 1\right\} \geq 1 - \delta, \tag{34}$$

where $C_1 = 8/\log(1 + \sigma_n^2)$. Observe that Equation 34 implies that with probability of at least $1 - \delta$ the cumulative regret is sublinear, i.e. $\lim_{T\to\infty} R_T/T = 0$. Now we complete the proof via contradiction. Consider any $\varepsilon > 0$. Suppose that with probability greater than $\delta$, $\lim_{T\to\infty} Q((s, ha, y), \xi_T) \geq Q((s, ha, y), \xi^*) + \epsilon$. If this is the case, then with probability greater than $\delta$ the regret will be linear with $T$, contradicting Theorem 3 from [41]. This completes the proof that for any $\epsilon > 0$, $\lim_{T\to\infty} Q((s, ha, y), \xi_T) < Q((s, ha, y), \xi^*) + \epsilon$ with probability of at least $1 - \delta$. $\square$

In our experiments, we used $c_{bo} = 2$ as described in Section 5. The GP parameters that used in the experiments are described in Section D.2.

## D  Additional Experiment Details

### D.1  Autonomous Car Navigation Domain

An autonomous car must navigate along roads to a destination as illustrated in Figure 1 by choosing from actions *{up, down, left, right}*. There are four types of roads: *{highway, main road, street, lane}* with different transition duration characteristics. The transition duration distribution for each type of road is unknown. Instead, the prior belief over the transition duration distribution for each road type is modelled by a Dirichlet distribution. Thus, the belief is represented by four Dirichlet distributions. Upon traversing each section of road, the agent receives reward $-time$ where $time$ is the transition duration for traversing that section of road. Upon reaching the destination, the agent receives a reward of 80.

For each type of road, it is assumed that there are three possible outcomes for the transition to the next junction, $\{fast, medium, slow\}$. For each type of road, the prior parameters for the Dirichlet distribution are $\{\alpha_{fast} = 1, \alpha_{medium} = 1, \alpha_{slow} = 0.4\}$. The transition duration associated with each outcome varies between each of the road types as follows:

- *highway*: $\{time_{fast} = 1, time_{medium} = 2, time_{slow} = 18\}$
- *main road*: $\{time_{fast} = 2, time_{medium} = 4, time_{slow} = 13\}$
- *street*: $\{time_{fast} = 4, time_{medium} = 5, time_{slow} = 11\}$
- *lane*: $\{time_{fast} = 7, time_{medium} = 7, time_{slow} = 8\}$

The prior over transition duration distributions implies a tradeoff between risk and expected reward. According to the prior, the *highway* road type is the fastest in expectation, but there is a risk of incurring very long durations. The *lane* road type is the slowest in expectation, but there is no possibility of very long durations. The *main road* and *street* types are in between these two extremes.

After executing a transition along a road and receiving reward $-time$, the agent updates the Dirichlet distribution associated with that road type, refining its belief over the transition duration distribution for that road type.

### D.2  Gaussian Process Details

We found that optimising the GP hyperparameters online at each node was overly computationally demanding. Instead, for all experiments we set the observation noise to $\sigma_n^2 = 1$ and the length scale to $l = \frac{1}{5y}$, where $y$ is the state factor representing the adversary budget. These parameters were found to work well empirically.

The prior mean for the GP was set to zero, which is optimistic for the minimising adversary. The Shogun Machine Learning toolbox [40] was used for GP regression.

### D.3 Policy Gradient Method Details

The method *CVaR BAMDP PG* applies the CVaR gradient policy update from [45] with the vectorised belief representation for BAMDPs with linear function approximation proposed in ([15], Section 3.3). The function approximation enables generalisation between similar beliefs.

Following the belief representation proposed by [15], we draw a set $M$ of MDP samples from the root belief. For each sample, $T_m \in M$, we associate a particle weight $z_m(h)$. The vector $z(h)$ containing the particle weights is a finite-dimensional approximate representation of the belief. The weights are initialised to be $z_m(h_0) = \frac{1}{|M|}$. During each simulation, as transitions are observed the particle weights are updated, $z_m(has') \propto T_m(s, a, s')z_m(h)$. We combine the belief feature vector, $z(h)$, with a feature vector representing the state-action pair, $\phi(s, a)$, in a bilinear form

$$F(h, s, a, \mathbf{W}) = z(h)^T \mathbf{W} \phi(s, a) \tag{35}$$

where $\mathbf{W}$ is a learnable parameter matrix. As we address problems with finite state and action spaces, for the state-action feature vector, $\phi(s, a)$, we chose to use a binary feature for each state-action pair. The action weighting, $F(h, s, a, \mathbf{W})$ defines a softmax stochastic action selection policy

$$f(a|h, s, \mathbf{W}) = \frac{\exp(F(h, s, a, \mathbf{W}))}{\sum_{a'} \exp(F(h, s, a', \mathbf{W}))} \tag{36}$$

where $f(a|h, s, \mathbf{W})$ is the probability of choosing action $a$ at history $h$, state $s$, and current parameter matrix $\mathbf{W}$. To perform a simulation, we sample a model from the prior belief using root sampling [15] and simulate an episode by choosing actions according to $f(a|h, s, \mathbf{W})$.

After performing a minibatch of simulations, we update the parameter matrix

$$\mathbf{W} \leftarrow \mathbf{W} + \lambda \nabla_{\text{CVaR}} \tag{37}$$

where $\nabla_{\text{CVaR}}$ is the Monte Carlo estimate of the CVaR gradient from ([45], Equation 6) and $\lambda$ is the learning rate.

We use $|M| = 25$, as this number of particles performed well in [15]. Following [45], we used a minibatch size of 1000 simulations for each update. To initialise the parameter matrix, we first computed the policy for the *CVaR VI Expected MDP* method. We denote by $\mathbf{W}(s, a, m)$ the parameter matrix entry associated with $s, a, m$. We set $\mathbf{W}(s, a, m) = 2$ for all $m$ if $a$ is chosen at $s$ by *CVaR VI Expected MDP*, and $\mathbf{W}(s, a, m) = 1$ for all $m$ otherwise.

## E  Additional Results

### E.1  Comparison with Random Action Expansion

Table 2 includes results for *RA-BAMCP* when the actions for the adversary are expanded randomly, rather than using Bayesian optimisation. For these results with random action expansions, we used double the number of trials that were used with Bayesian optimisation: 200,000 trials at the root node, and 50,000 trials per step thereafter. This means that the total computational resources used with and without Bayesian optimisation was comparable. Even with a comparable amount of computation time, *RA-BAMCP* with random action expansions performed worse than *RA-BAMCP* using Bayesian optimisation to select actions to expand.

| Method | Time (s) | $\text{CVaR}_{0.03}$ | $\text{CVaR}_{0.2}$ | Expected Value |
|---|---|---|---|---|
| *RA-BAMCP* (Random expansions, $\alpha = 0.03$) | 29.2 (0.1) | 8.24 (0.29) | 9.94 (0.06) | 10.73 (0.05) |
| *RA-BAMCP* (Random expansions, $\alpha = 0.2$) | 72.6 (0.1) | 0.0 (0.0) | 18.39 (0.99) | 54.29 (0.53) |

(a) Betting Game domain

| Method | Time (s) | $\text{CVaR}_{0.03}$ | $\text{CVaR}_{0.2}$ | Expected Value |
|---|---|---|---|---|
| *RA-BAMCP* (Random expansions, $\alpha = 0.03$) | 214.7 (0.1) | 16.06 (0.35) | 24.63 (0.25) | 36.92 (0.20) |
| *RA-BAMCP* (Random expansions, $\alpha = 0.2$) | 221.5 (0.2) | 12.05 (0.46) | 26.84 (0.38) | 43.15 (0.26) |

(b) Autonomous Car Navigation domain

Table 2: Results from evaluating the returns over 2000 episodes for *RA-BAMCP* using random action expansions for the adversary. Brackets indicate standard error of the mean. Time indicates average total computation time per episode to perform simulations.

## E.2 Return Distribution in Betting Game

The histograms in Figure 3 show the return distributions for *RA-BAMCP* and *BAMCP* in the Betting Game domain.

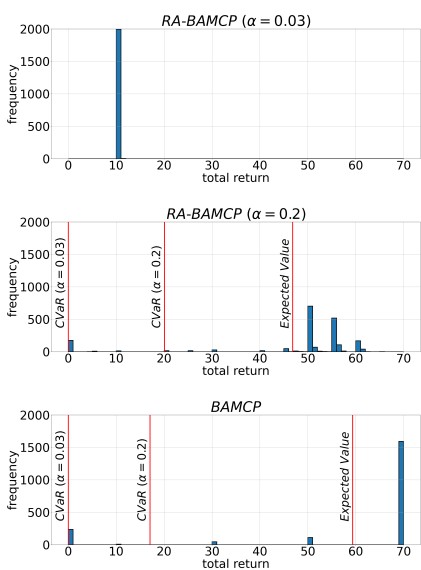

Figure 3: Histogram of returns received in the Betting Game domain.

## E.3 Hyperparameter Tuning for Policy Gradient Method

Table 3 presents results for *CVaR BAMDP PG* after $2 \times 10^6$ training simulations using a range of learning rates. A learning rate of $0.001$ performed the best across both domains and $\alpha$ values that we tested on. A learning rate of $\lambda = 0.01$ performed worse on the betting game with $\alpha = 0.2$, and a learning rate of $\lambda = 0.0001$ performed worse across both domains and confidence levels. Therefore, the results presented in the main body of the paper use a learning rate of $\lambda = 0.001$. Training curves using this learning rate can be found in Section E.4.

| Method | $\text{CVaR}_{0.03}$ | $\text{CVaR}_{0.2}$ | Expected Value |
|---|---|---|---|
| *CVaR BAMDP PG* ($\lambda = 0.01, \alpha = 0.03$) | 3.00 (0.0) | 10.86 (0.33) | 27.31 (0.24) |
| *CVaR BAMDP PG* ($\lambda = 0.001, \alpha = 0.03$) | 2.90 (0.06) | 10.57 (0.33) | 27.34 (0.24) |
| *CVaR BAMDP PG* ($\lambda = 0.0001, \alpha = 0.03$) | 2.31 (0.14) | 8.74 (0.28) | 26.19 (0.25) |
| | | | |
| *CVaR BAMDP PG* ($\lambda = 0.01, \alpha = 0.2$) | 0.0 (0.0) | 18.03 (0.91) | 35.73 (0.27) |
| *CVaR BAMDP PG* ($\lambda = 0.001, \alpha = 0.2$) | 0.00 (0.0) | 19.85 (0.97) | 46.65 (0.37) |
| *CVaR BAMDP PG* ($\lambda = 0.0001, \alpha = 0.2$) | 0.0 (0.0) | 14.72 (0.76) | 43.14 (0.53) |

(a) Betting Game domain

| Method | $\text{CVaR}_{0.03}$ | $\text{CVaR}_{0.2}$ | Expected Value |
|---|---|---|---|
| *CVaR BAMDP PG* (Learning rate= $0.01, \alpha = 0.03$) | 24.0 (0.0) | 25.02 (0.04) | 28.97 (0.05) |
| *CVaR BAMDP PG* (Learning rate= $0.001, \alpha = 0.03$) | 23.92 (0.08) | 25.38 (0.05) | 28.97 (0.05) |
| *CVaR BAMDP PG* (Learning rate= $0.0001, \alpha = 0.03$) | 20.11 (2.12) | 24.86 (0.34) | 29.00 (0.09) |
| | | | |
| *CVaR BAMDP PG* (Learning rate= $0.01, \alpha = 0.2$) | 3.7 (0.97) | 24.88 (0.60) | 51.78 (0.36) |
| *CVaR BAMDP PG* (Learning rate= $0.001, \alpha = 0.2$) | 10.05 (0.74) | 24.70 (0.39) | 49.67 (0.36) |
| *CVaR BAMDP PG* (Learning rate= $0.0001, \alpha = 0.2$) | -13.25 (2.78) | 19.94 (0.91) | 51.19 (0.42) |

(b) Autonomous Car Navigation domain

Table 3: Results from evaluating the returns over 2000 episodes for *CVaR BAMDP PG* after training for $2 \times 10^6$ simulations using different learning rates. Brackets indicate standard error of the mean.

### E.4 Training Curves for Policy Gradient Method

The training curves presented Figures 4-7 illustrate the performance of the policy throughout training using the policy gradient approach, with the learning rate set to $\lambda = 0.001$. To generate the curves, after every 20,000 training simulations the policy is executed for 2,000 episodes and the CVaR is evaluated based on these episodes.

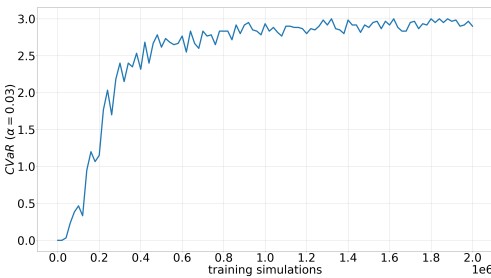

Figure 4: Training curve for policy gradient optimisation in the betting game domain with $\alpha = 0.03$.

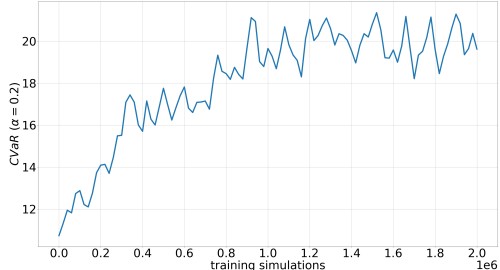

Figure 5: Training curve for policy gradient optimisation in the betting game domain with $\alpha = 0.2$.

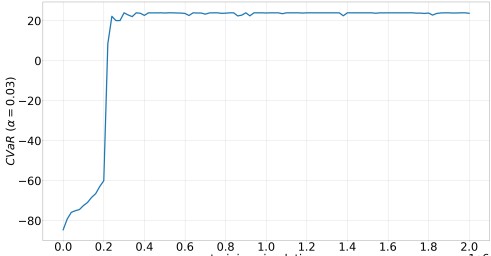

Figure 6: Training curve for policy gradient optimisation in the navigation domain with $\alpha = 0.03$.

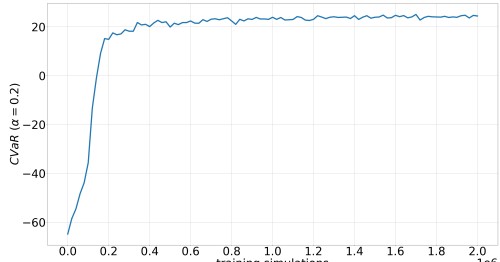

Figure 7: Training curve for policy gradient optimisation in the navigation domain with $\alpha = 0.2$.