# OpenReview forum: "Risk-Averse Bayes-Adaptive Reinforcement Learning"
_NeurIPS.cc/2021/Conference — NeurIPS 2021 Poster_

### Official Review · Reviewer_4TDb · 2021-07-08

**Rating:** 6
**Confidence:** 4

**Summary:**

The paper discusses parametric (epistmic) versus internal (aleatory) uncertainty in the context of Bayesian reinforcement learning. It extends previous methods for CVaR optimisation in MDPs to the BAMDP setting.

The response clarified some minor issues, but my main points remain unchanged.

**Limitations And Societal Impact:**

There is no clear distinction between aleatory and epistemic uncertainty. This is important, as a decision maker might be risk averse with respect to one type of risk, but not the other. The current approach gives no way to control the behaviour.


**Main Review:**

The paper discusses parametric (epistmic) versus internal (aleatory) uncertainty in the context of Bayesian reinforcement learning.  Although it claims a significant amount of novelty, a number of works [1,2,3] already combine both types of uncertainty in risk-sensitive Bayesian reinforcement learning, There have also been works on risk-averse tree search [4]. The technical novelty is also limited, as it directly uses earlier results that extended dynamic-programming techniques for aleatory CVaR risk from MDPs to epistemic uncertainty. However, this is the first paper that considers risk-aversion with the Bayes-adaptive MDP framework. The results show that this is a viable methohd.


I find it hard to understand how the Bayesian framework fits here exactly. In particular, since the budget is shared arbitrarily between prior and transition perturbations, this implies that there is no clear distinction between aleatory and epistemic uncertainty. This is important, as a decision maker might be risk averse with respect to one type of risk, but not the other. The current approach gives no way to control the behaviour.

Overall, I find the writing quite opaque compared to the Chow et al paper.

Proposition 2 $G_{\pi, \sigma}$ is undefined. I assume it is the adversary strategy.


Minor comments:

> Line 87 "In this paper, we interpret Z as a reward to be maximised."

Do you mean return?

> Line 186 "all perturbations over T as Xi = x_T_p in T^Xi_p"

This is confusing.


General questions:

Would be very interesting to see experiments on varying the horizon length.
Would perhaps CVaR VI BAMDP outperform RA-BAMCP even on the large setting if you
used a smaller horizon?


[1] Depeweg et al. 2018, "Decomposition of uncertainty in Bayesian deep learning for efficient and risk-sensitive learning."
[2] Clements et al, 2019, "Estimating risk and uncertainty in deep reinforcement learning."
[3] Eriksson et al, 2021, "SENTINEL: Taming Uncertainty with Ensemble based Distributional Reinforcement Learning"
[4] Lecarpentier and Rachelsson, 2019, "Non-Stationary Markov Decision Processes a Worst-Case Approach using Model-Based Reinforcement Learning"


**Time Spent Reviewing:**

3

---

> ### Author Response · Authors · 2021-08-09
> **Response to reviewer 4TDb**
>
> Thank you for the valuable time you have spent reviewing our submission.
>
> **Comparison to related work:**
>
> Thank you for the references on uncertainty in reinforcement learning, we will add these to the discussion in the related work section. As noted, ours is the first work to use the Bayes-adaptive MDP framework to consider aversion to both aleatoric and epistemic risk.
>
> **Distinction between epistemic and aleatoric risk:**
>
> In our work, the objective we optimise is the CVaR of the total return, where any run is subject to both epistemic and aleatoric uncertainty. This means that our algorithm works to mitigate both forms of uncertainty in such a way that the worst 100$\alpha$% of runs is optimised. This is regardless of what proportion of the risk in the worst 100$\alpha$% of runs is attributed to aleatoric or epistemic uncertainty.  Assuming that the decision-maker wishes to avoid the risk of poor returns, our formulation is a more intuitive way to achieve this desired behaviour than choosing a scalar weighting that penalises each form of uncertainty.
>
> In our experiments we evaluate each method in the presence of both aleatoric and epistemic uncertainty. We compare against a risk-neutral algorithm (BAMCP), and an algorithm which only mitigates the aleatoric uncertainty (CVaR VI Expected MDP). Our results show that CVaR VI Expected MDP outperforms the risk-neutral algorithm, indicating that only mitigating the aleatoric uncertainty improves performance in the Bayes-adaptive setting we address. However, our algorithm which mitigates both sources of uncertainty outperforms CVaR VI Expected MDP which indicates that considering both sources of uncertainty is crucial for optimal risk-sensitive behaviour in the presence of both forms of uncertainty.
>
> $G_{\pi, \sigma}$ notation:
>
> This is the return distribution induced by the agent and adversary strategies. This notation is defined on Line 139. We use $\sigma$ to denote the adversary strategy, which is defined on Line 138.
>
> **Line 87:**
>
> We will correct this to say “we interpret Z as the return to be maximised”. Thank you for pointing out this mistake.
>
> **CVaR VI BAMDP performance for small horizons:**
>
> CVaR VI BAMDP performs dynamic programming over the entire history-dependent state space, so this method gives an accurate solution to the Bayes-adaptive CVaR optimisation problem we address in this work. However, it is not scalable to longer horizons as it requires value iteration over the entire history-dependent state space which grows exponentially. For a short horizon, we would expect the solution for CVaR VI BAMDP to perform well, and therefore this approach may outperform RA-BAMCP for small horizons.

---

### Official Review · Reviewer_hcVR · 2021-07-15

**Rating:** 4
**Confidence:** 3

**Summary:**

The paper proposes a method to optimize the conditional value-at-risk of the total return in Bayes-adaptive MDPs. They formulate the problem as a two player stochastic game and develop a Monte Carlo tree search based approximate algorithm. They show that the optimized policy is risk-averse to both epistemic/model and aleatory/transition uncertainties. They report several experiments to show the utility of their proposed method.

**Limitations And Societal Impact:**

limitations and impacts are briefly addressed.

**Main Review:**

It is nice to have an illustrative example to clarify ideas at the beginning of the paper, but is the example presented in introduction useful for the purpose? Model uncertainty refers to the uncertainty associated to not knowing the transition probabilities. And the transition uncertainty refers to the inherent stochasticity. It is not clear how the situations presented in the example correspond to those uncertainties. It rather makes things more confusing.

In line 121, V is defined as \min_\pi CVAR(G), where G is defined in terms of reward, not cost. Should this be maximised instead, as shown in equation (4)?

What is the definition of a node? Do you aggregate/discretize to define a state and/or node? What is the impact of such process?

RA-BAMCP is an online algorithm which involves computing posteriors at each node of the tree in realtime. Using Beta/Dirichlet distribution, as done in the paper, provides a closed form approach to compute posteriors. But what happens when a closed form approach is not avaialable for a distribution? Does it scale? How about some empirical analysis?

One main claim in the paper is to apply CVaR over both the model and transition uncertainties. What difference this makes compared to general robust and/or risk averse methods studied in the literature? This is an important question to address, both theoretically and empirically. The paper misses this point. The empirical evaluation may show the difference more clearly by addressing only model uncertainty, only transition uncertainty and then addressing both types of uncertainties. Also, it can be interesting to compare against general robust, soft-robust and risk-averse methods available in the literature.

As the paper uses many different concepts like SG, MCTS, OFU/UCRL, GP and so on, the paper feels highly convoluted to read. In my opinion, the paper is not well written. A reorganization of the contents focusing readability and clarity may help.

**Time Spent Reviewing:**

7

---

> ### Author Response · Authors · 2021-08-09
> **Response to reviewer hcVR**
>
> Thank you for the valuable time you have spent reviewing our submission.
>
> **Illustrative example:**
>
> In the illustrative example, there is transition certainty corresponding to the distribution over transition times. For example, on a day with normal traffic this could be model1 = {prob(1 minute) = 0.5, prob (2 minutes) = 0.4, prob(3 minutes) = 0.1}. This transition uncertainty might be caused by traffic lights, pedestrian crossings, or other sources of inherent stochasticity. On a day with busy traffic, the transition distribution might be model2 = {prob(1 minute) = 0.2, prob(2 minutes) = 0.4, prob(3 minutes) = 0.4}. The model uncertainty in our illustrative example refers to the fact that at the start of each episode we do not know the traffic conditions, and therefore do not know which transition distribution is the appropriate one. In our experiments we use Dirichlet distributions to model the uncertainty over the transition duration distributions.
>
> **Line 121:**
>
> You are correct, this should be max not min. We will fix this mistake. Thank you for pointing this out.
>
> **Definition of a node:**
>
> Each node in the search tree corresponds to a unique history of states, agent actions, and adversary perturbations. The progressive widening procedure is used to discretise adversary actions because the adversary actions are continuous. When combined with Bayesian optimisation, this process ensures that we focus search on promising adversary actions. Our experiments show that this Bayesian optimisation procedure improves the efficiency of our algorithm.
>
> **Scalability for complex posteriors:**
>
> Improving scalability to more complicated posteriors is a subject for future work, and something which we mention in the section on limitations. One potential approach to handling more complicated posteriors which we wish to explore is to approximate the posterior using a particle filter.
>
> **Reasons for applying CVaR to both model and transition uncertainties:**
>
> Our approach is appropriate for domains where the risk of obtaining a poor return on any episode is to be avoided. Approaches which only address model uncertainty typically ensure that the expected return is high for all possible models. However, they do not consider the fact that the returns under any model may have high variance (i.e. high transition uncertainty) and therefore there may still be a large chance for a poor return for any particular episode.
>
> Approaches which only address transition uncertainty assume that the model/simulator is accurate, and do not consider the risk of poor performance if there are inaccuracies in the model/simulator.
>
> In the experiments we evaluate the performance of each algorithm in the presence of both model and transition uncertainty. We compare against a risk-neutral algorithm (BAMCP), and algorithm which only mitigates the transition uncertainty (CVaR VI Expected MDP). Our results show that CVaR VI Expected MDP outperforms the risk-neutral BAMCP, indicating that only mitigating the transition uncertainty improves risk-sensitive performance. However, our algorithm which mitigates both sources of uncertainty outperforms CVaR VI Expected MDP which indicates that considering both sources of uncertainty is important to strong performance in the presence of both model and transition uncertainty.

---

### Official Review · Reviewer_aH4a · 2021-07-16

**Rating:** 5
**Confidence:** 5

**Summary:**

The authors propose a new method introducing a risk-averse Bayes-adaptive reinforcement learning. This method aims at tackling two types of uncertainty that have mostly been tackled separately in former works: the epistemic uncertainty coming from imperfect knowledge about the underlying system and aleatoric uncertainty stemming from the internal stochasticity of the system. The authors formulate the risk-averse optimisation problem as a CVaR maximisation of the return in model-based Bayesian RL, which they show is equivalent to solving a stochastic game.  The stochastic game view shows that the proposed method effectively tackles both types of uncertainty and leads to the design of an algorithm based on Bayesian optimisation and MCTS to address the problem.  The proposed algorithm is evaluated against several baselines, demonstrating superior performances on two different tasks.

**Ethical Concerns:**

I don't foresee ethical issues. However, social impacts have not been considered.

**Limitations And Societal Impact:**

Limitations of the methods have been provided. However, social impacts were not discussed and answered as No. I am not sure but I think safety can play a critical role in social impact especially in navigation tasks for example. Maybe the authors could look a bit more into this.

**Main Review:**

In general, I found the paper an interesting read, well organised and provides a promising approach to risk-averse optimisation problems. I have still some questions notably regarding how practical the approach could be beyond the provided small-scaled experiments which still shows interesting results:

1) In related work sections you mention policy gradient methods that assume the agent has access to the true underlying MDP for training/planning in contrast with your method for which you "only have access to a prior belief distribution over MDPs". Though in your experiments you say that "For each episode, the true underlying MDP was sampled from the prior distribution".  Can you please help me understand if this boils to having access to a true MDP (making the problem formulation less novel)?

3) In the BayesOptPerturbation what motivated the use of an RBF kernel over, for instance, a matern-5/2 kernel? Are there standard assumptions or beliefs on the smoothness of the underlying black-box optimised with BO? Do the authors assume stationarity? Can you please help me understand what specific assumptions are made on the function when using BO?

4) Results of Proposition 3 are interesting. How can those results be considered to arrive at a convergence bound of the overall proposed method?

5) When it comes to the experiment, the main concern is in terms of scalability with respect to the number of states, actions and horizons. As one of the claims of the paper is that the proposed method tackles both aleatoric and stochastic uncertainty, it could be interesting to present a set of experiments with variable degrees of aleatoric/stochastic uncertainty to see what are the strengths and limitations of the method.  Can the authors run such experiments and present an updated version analysing the effect of those degrees?

6) There are also references that just point to the appendix as a whole. Providing a more accurate reference could definitely aid in readability.

Related Work:
There are some related works that could be cited from both the Bayesian optimisation and model-based RL that could strengthen the positioning of the paper:
1. Scalable Global Optimization via Local Bayesian Optimization
2. SAMBA: Safe Model-Based & Active Reinforcement Learning
3. An Empirical Study of Assumptions in Bayesian Optimisation
4. Taking the Human Out of the Loop: A Review of Bayesian Optimization
5. Practical Bayesian Optimization of Machine Learning Algorithms


**Time Spent Reviewing:**

5

---

> ### Author Response · Authors · 2021-08-09
> **Response to reviewer aH4a**
>
> Thank you for the valuable time you have spent reviewing our submission.
>
> **1. Problem formulation:**
>
> Existing policy gradient methods are allowed to execute many thousands of episodes in the true underlying MDP for training, and then test in the same MDP.
>
> In our approach, during search/planning time our approach only has access to the prior over MDPs and uses this to learn an adaptive, or history-dependent, policy. To test our approach, for each episode, we sample an MDP and execute one episode only in that MDP. The agent does not know what the true MDP is at any time. The only knowledge available at the start of the episode is the prior over MDPs. The Bayes-adaptive policy is only allowed to adapt or learn about the particular sampled MDP over the course of a single episode. A new MDP is then sampled for the next episode.
>
> Thus, the key difference is that our method must learn an adaptive policy which mitigates risk regardless of which (unknown) MDP has been sampled as the true underlying MDP for each episode. In other words, it must mitigate the epistemic uncertainty (due to a different MDP being sampled each episode) in addition to the aleatoric uncertainty (stochastic outcomes within the MDP). Existing policy gradient methods only optimise risk-sensitive performance in a single MDP, and therefore they only optimise for risks due to the aleatoric uncertainty.
>
> **2. Bayesian optimisation:**
>
> In the proof of Proposition 3 in the supplementary material, we prove that the underlying function to be optimised is continuous (i.e. smooth). Convergence results for Bayesian optimisation exist provided that the underlying function is continuous, and the kernel used is universal [1]. This allows us to prove that the Bayesian optimisation procedure is guaranteed to converge provided that we use a universal kernel (such as RBF, Matern, etc.). Thus, we decided to use the RBF kernel as it is a universal kernel, but in future work we could also explore using other universal kernels such as the Matern kernel.
>
> In the proof of Proposition 3, we make the assumption that after a new perturbation is added, the value of this perturbation is estimated using MCTS to within $\sigma_n$ of the true optimal value. This value estimate is added to the GP. To use the theoretical results from [1] the proof of Proposition 3 assumes that we do not update existing value estimates in the GP after subsequent perturbations are added and more trials of MCTS. We will clarify this in the proof of Proposition 3.
>
> **3. Convergence bound:**
>
> Analysing the convergence of the full MCTS algorithm is something we wish to address in future work.
>
> **4. Experiments analysing effect of each type of uncertainty:**
>
> In the experiments we compare against a risk-neutral algorithm (BAMCP), and algorithm which only mitigates the aleatoric uncertainty (CVaR VI Expected MDP). Our results show that CVaR VI Expected MDP outperforms the risk-neutral BAMCP, indicating that only mitigating the aleatoric uncertainty improves risk-sensitive performance in our problem setting. However, our algorithm which mitigates both sources of uncertainty outperforms CVaR VI Expected MDP which indicates that considering both sources of uncertainty is important to strong performance in the presence of both sources of uncertainty.
>
> Unfortunately, the conference rules prohibit us from submitting a revised version of the paper with additional experiments. However, in future work we would like to test our approach on experimental domains which more clearly separate the effect of each source of uncertainty.
>
> **5. References to appendix:**
>
> We will edit the paper to refer to more specific parts of the appendix.
>
> **Related work:**
>
> Thank you for the suggestions concerning the related work. We will add a discussion of these to the related work section.
>
> [1] Srinivas, Niranjan, et al. "Gaussian process optimization in the bandit setting: No regret and experimental design." ICML 2010

---

### Official Review · Reviewer_Kn8T · 2021-07-16

**Rating:** 7
**Confidence:** 4

**Summary:**

A method for risk-averse Bayes-adaptive RL is developed theoretically and an implementation is provided, together with practical approximations for feasibility. The MDP's transition parameters $\theta$ are a random variable (hence Bayes-adaptive), leading to two types of relevant uncertainty: that over the MDP itself (parametric, epistemic) and the inherent uncertainty of the MDP transition (internal, aleatoric). The focus of the paper is on optimizing for risk, here through the Conditional Value at Risk (CVaR) metric, w.r.t. both uncertainty types at once.

Inspired by [9], the authors propose a seemingly novel derivation for CVaR (Proposition 1), adapted to the Bayesian treatment of $\theta$. The so-formulated CVaR is then used to pose a two-player stochastic game problem [9, 33], with a new structure. Optimization is done with MC tree search, somewhat similar to [33], but significantly extended. The adversary needs to perturb both the distribution over the transition with parameters $\theta$  as well as the individual transition probabilities (targeting the risk envelope under all uncertainty), and the search is heavier due to necessary posterior updates (Bayesian $\theta$). For feasibility, the continuous space of possible perturbation actions of the adversary is approximated through a discrete set, using progressive widening with Bayesian optimization to further extend it during search.

The method is validated on two toy scenarios, with adequate performance compared to other baselines.

**References**

[9] Yinlam Chow, Aviv Tamar, Shie Mannor, and Marco Pavone. Risk-sensitive and robust decision-making: a CVaR optimization approach. NIPS 2015.

[15] Arthur Guez, David Silver, and Peter Dayan. Efficient Bayes-adaptive reinforcement learning using sample-based search. NIPS 2012.

[33] Apoorva Sharma, James Harrison, Matthew Tsao, and Marco Pavone. Robust and adaptive planning under model uncertainty. ICAPS 2019.


**Limitations And Societal Impact:**

Limitations are adequately addressed. The presented work is generic, societal impact should not be directly relevant in this case.

**Main Review:**

**Originality**

Optimizing CVaR under both epistemic and aleatoric uncertainty has not been explored before, as far as I am aware. The perspective on CVaR in *Proposition 1* appears novel to me, it is inspired by [9], with the addition of adversarial minimization of the value function w.r.t. the distribution over transitions (which is unique to the Bayes-adaptive case). The combination of the chosen strategies (MCTS + progressive widening + Bayesian optimization) is also new to me. Related work appears adequately cited, although I think the claim that this is the first use of MCTS for CVaR optimization in sequential decision making might be overstated ([33] already does this, but only for epistemic uncertainty).

**Strengths**

- The proposed approach appears technically sound. The theoretical analysis is logical, extending the ideas of the cited previous works [9,15,33]. Given the technical complexity of the topic at hand, I believe this is valuable.
- In terms of clarity, the paper is constructed well, motivation is presented in a reasonable order. The provided code in the supplementary material is appreciated, and should further help with understanding.
- Full Bayesian modeling of risk in the context of RL is a very significant topic with broad applicability (finance, robotics, safety-critical systems, etc.). I believe it deserves more attention, and the paper does pinpoint specific scalability issues in the fully-Bayesian setting that future research should aim to address.
- To an extent, the proposed approach does manage to address one of the scalability problems listed below (the prohibitively large continuous perturbation space of the adversary) through the discrete perturbation action set + BayesOpt (GP, Gaussian kernel) combination. This makes the method faster than the relevant baselines, considering both epistemic and aleatoric uncertainty. Still, performance needs to be improved further and it remains an open research question whether such approximations can work for more realistic problems.

**Weaknesses**

- The main weakness of the proposed solution is its scalability (exponential state space, heavy tree search operations due to posterior updates, limited exhaustion of adversary actions in an otherwise continuous high-dimensional space, no MCTS root sampling possible). I believe this is sufficiently acknowledged by the authors in the last two sections of the paper. It is not my expectations that all of these issues should be addressed in a single paper. Be it as it may, in its current version the approach is likely not applicable to real-world scenarios.
- I also find that the experimental section could be improved. The two presented settings are very limited. This is understandable given the above, but I would have preferred a more varied set of environments with larger state spaces, even if that means evaluating only the proposed approach on some of them. Otherwise it is hard to judge the degree of generalization. The comparison to *CVaR VI Expected MDP*, which averages over the $\theta$ distribution, is appreciated, but I would have liked to see the related *RAMCP* [33] in the comparison for symmetry, as well as an extended analysis of the modeling choices & structure of the proposed method (e.g. how the assumed prior entropy relates to the overall result / runtime budget).



Overall, I feel there is merit to the presented theoretical framework and the paper does a decent job at highlighting some of the core issues that remain to be solved, even if I find the experimental section somewhat lacking.


**Time Spent Reviewing:**

8

---

> ### Author Response · Authors · 2021-08-09
> **Response to reviewer Kn8T**
>
> Thank you for the valuable time you have spent reviewing our submission.
>
> **Novelty of MCTS for CVaR optimisation in comparison to [33]:**
>
> We will modify the claim to this being the first work to use MCTS to optimise CVaR in the presence of aleatoric uncertainty.
>
> **Scalability and experiments on larger domains:**
>
> We acknowledge that the scalability of the approach is limited, as discussed in the limitations section. In future work, we wish to improve the scalability of our approach through the utilisation of function approximation (e.g. deep learning) to generalise between similar states and posteriors. This improved scalability will enable us to test our approach on larger domains.
>
> In the supplementary material we will include an additional experiment analysing the performance of our approach for varying computational budget.

---

### Official Review · Reviewer_JA6Z · 2021-07-17

**Rating:** 6
**Confidence:** 3

**Summary:**

This paper considers the problem of risk-averse problems in Bayesian RL inspired by real-world scenarios, aiming to address the two sources of this uncertainty: the parametric (or epistemic) uncertainty and internal (or aleatoric) uncertainty. This paper applies CVaR as the risk measurement and optimizes it by formulating it as a stochastic game, which is novel, and solves it with MCTS.

**Limitations And Societal Impact:**

Yes

**Main Review:**

The problem discussed in this paper is important. This paper is well-written. The methods used in this paper are well-motivated. The main framework is built on top of optimizing the CVaR over the return of BAMDP. The Bayes-Adaptive Stochastic Game formulation is new and I think maybe the authors were inspired by adversarial RL methods.

There are some limitations and questions:

1. Why optimizing CVaR? The Mean-Variance [1] is also a direction. Can you tell me the reason?
2. Can you also compare your method with VariBAD [2]?
3. In lines 40-41, “By optimising the CVaR of the return in the BAMDP, our approach simultaneously addresses parametric and internal uncertainty under a single framework.”, is not clear in the paper, can you elaborate it?
4. You proposed Bayes-Adaptive Stochastic Game, can you discuss more about the differences between your problem and adversarial RL methods [3]?
5. In lines 232-235, can you give a clearer explanation? Why optimizing CVaR is hard than the expected value? To what magnitude your searching method addresses the growing set of possible histories compared with other methods?
6. In your experiments, you used many simple scenarios. As you aim to address the two uncertainties in BADMP with risk-averse policies, can you show more complex scenarios, for example,  CARLA and Torcs or other scenarios, to demonstrate more risk-averse behaviour?

[1] Mean-Variance Policy Iteration for Risk-Averse Reinforcement Learning.
[2] VariBAD: A Very Good Method for Bayes-Adaptive Deep RL via Meta-Learning
[3] Robust Adversarial Reinforcement Learning


**Time Spent Reviewing:**

3 hours

---

> ### Author Response · Authors · 2021-08-09
> **Response to reviewer JA6Z**
>
> Thank you for the valuable time you have spent reviewing our submission.
>
> **1. CVaR versus mean-variance:**
>
> CVaR is a coherent risk metric [1], meaning that it satisfies common sense axioms that a sensible risk measure should satisfy. The mean-variance criterion is not coherent as it fails to satisfy some of the axioms for coherency.
>
> We will illustrate one of the issues with the mean-variance criterion with an example. Consider a problem where there are 3 possible cases, or outcomes, which occur with probability 1/3 each. For policy $\pi_1$, the returns for each case are: {case1: 10, case2: 10, case3: 15}. For policy $\pi_2$ the returns for each case are: {case1: 11, case2: 11: case3: 20}.
>
> Clearly, $\pi_2$ always attains the better return for any case, so any rational decision maker should prefer policy $\pi_2$. Indeed, optimising CVaR for any confidence level will always prefer policy $\pi_2$.
>
> However, let's now consider the mean-variance objective: mean - $\beta$ * variance. For simplicity, we will set $\beta$ to 1. The mean-variance objective for $\pi_1$ is 6.1, while the mean-variance criteria for $\pi_2$ is -4. Therefore, the mean-variance objective prefers $\pi_1$, which is irrational as $\pi_1$ performs worse in all cases. A more thorough discussion of the issues with non-coherent risk measures is in [2].
>
> **2. Comparison with VariBAD:**
>
> VariBAD is an approach which utilises deep learning techniques to learn to optimise expected value in the Bayes-Adaptive reinforcement learning problem. VariBAD does not consider risk-sensitivity, which is the core focus of this work.
>
> **3. Parametric and internal uncertainty under a single framework:**
>
> We optimise CVaR in the Bayes-adaptive MDP setting. In the Bayes-adaptive setting, we assume that we have access to a prior over the true underlying MDP. To optimise CVaR in this setting (i.e. the worst 100$\alpha$% of runs), we must take into consideration that the outcome for any run is uncertain because of two sources of uncertainty: the true MDP model is unknown due to the uncertainty in the prior (parametric uncertainty), and the agent is subject to stochastic transitions within that true model (internal uncertainty).
>
> Proposition 1 illustrates formally that by optimising CVaR in the Bayes-Adaptive setting (the single framework that we present), the solution must mitigate both the risk of having a “bad” true underlying MDP, or having “bad” stochastic transitions occurring during the run. In other words, any policy which optimises CVaR in the Bayes-adaptive MDP must optimally avoid risks due to both parametric and internal uncertainty.
>
> **4. Comparison to Robust Adversarial RL:**
>
> In robust adversarial reinforcement learning (RARL), the adversarial agent is allowed to perturb the simulator. This results in more robust behaviour from the agent. One key difference is that in RARL it is unclear how much perturbing “power” should be given to the adversary to achieve the desired level of robustness in the agent. In other words, it is not clear how to choose the magnitude of the disturbing forces that the adversary is allowed to apply. It is also not clear what the agent ends up optimising in response to these disturbances.
>
> In our model-based stochastic game formulation, the action space of the adversary is defined precisely so that in response the agent’s optimal policy optimises CVaR.  Unlike RARL, the resulting optimisation objective is clear, and easy for the user to interpret. The desired level of risk-sensitivity can easily be tuned by changing the CVaR confidence level.
>
> In addition, we address the Bayes-adaptive setting, where the agent must find an adaptive policy which works under a prior over the MDP. RARL addresses the standard MDP setting. We will add a comparison to RARL in the related work.
>
> **5. Lines 232-235 - MCTS for search over histories, and difficulty of optimising CVaR:**
>
> We address the Bayes-adaptive MDP setting, where we only have access to a prior over the MDP. In this setting, we attempt to find a history-dependent policy. This means that the policy depends on the history observed up to the current state. This allows the policy to adapt to the posterior distribution over the MDP given the transitions observed so far during the current episode. However, the number of possible histories that the agent may encounter grows exponentially with the horizon. This means it is infeasible to compute the optimal policy at every possible history, using for example, value iteration.
>
> Our MCTS method addresses the exponential growth of possible histories by focusing search and computation on promising histories. This is inspired by existing work on Bayes-adaptive MDPs which uses MCTS to deal with the state space explosion and focus search in promising areas [6].
>
> The difficulty of optimising CVaR in MDPs is twofold. First, in the static risk setting that we address, the optimal CVaR policy may be history-dependent [3]. As noted above, we search over history-dependent policies as this is also necessary to address the Bayes-adaptive component of our problem setting. Second, CVaR optimisation algorithms require the introduction and optimisation of additional continuous variables. In dynamic programming algorithms this may involve optimising a continuous variable in an outer-loop [3], or augmenting the state-space with a continuous state factor [4]. In policy-gradient methods, a Lagrangian reformulation is often used, and an additional continuous variable is optimised on a slower time-scale than the policy parameters [5]. In our approach, this second difficulty is manifested in the additional continuous state factor, y, which we include in the state space.
>
> **6. Testing on more complex experimental domains:**
>
> Improving the scalability of our approach so that it may be applied to more complex domains is a subject of future work. In the conclusion we suggest that using function approximation (e.g. deep learning) to generalise between similar states and beliefs is a promising approach for greatly improving the scalability of our method.
>
> [1] Philippe Artzner, Freddy Delbaen, Jean-Marc Eber, and David Heath. Coherent measures of risk. Mathematical finance, 9(3):203–228, 1999.
>
> [2] Majumdar, Anirudha, and Marco Pavone. "How should a robot assess risk? Towards an axiomatic theory of risk in robotics." Robotics Research. Springer, Cham, 2020. 75-84.
>
> [3] Bäuerle, Nicole, and Jonathan Ott. "Markov decision processes with average-value-at-risk criteria." Mathematical Methods of Operations Research 74.3 (2011): 361-379.
>
> [4] Chow, Yinlam, et al. "Risk-sensitive and robust decision-making: a cvar optimization approach." NeurIPS (2015).
>
> [5] Chow, Yinlam, et al. "Risk-constrained reinforcement learning with percentile risk criteria." The Journal of Machine Learning Research 18.1 (2017): 6070-6120.
>
> [6] Guez, Arthur, David Silver, and Peter Dayan. "Efficient Bayes-adaptive reinforcement learning using sample-based search." Neural Information Processing Systems (2012).

---

### Decision · Program_Chairs · 2021-09-28

**Decision:**

Accept (Poster)

**Comment:**

This paper has been carefully discussed by the reviewers, the consensus is that while this paper applies CVaR to the fully-Bayesian setting and its motivation is appreciated (especially the motivation behind CVaR optimization in Bayesian MDP is to address the two sources of uncertainty: the parametric (or epistemic) uncertainty and internal (or aleatoric) uncertainty, which is good),  the contribution on extending CVaR MDPs onto the Bayes MDP setting is quite incremental. Similar to the standard MDP case, the authors managed to show a dual representation results for Bayes MDP and thus are able to modify algorithms such as MCTS to solve CVaR Bayes MDPs. Reviewers also mentioned concerns about scalability of the algorithm, and the lack of more realistic experiments (besides the small-scale experiments reported). For example some reviewers suggested more experimental details such as decoupling the effect of epistemic and aleatoric uncertainty on risk experimentally could also have been included, as well as considering an analysis of different priors (e.g. varying entropy, or an incorrect prior w.r.t. the true MDP distribution).

On the overall, the paper studied an interesting topic but its current form is below the acceptance threshold.




**Consistency Experiment:**

NeurIPS has a long history of experimentation. In 2014, NeurIPS ran an experiment in which 10% of submissions were reviewed by two independent committees to quantify the randomness in the review process. This year, we repeated a variant of this experiment to see how the quality of the review process has changed over time.  This paper was part of the experiment and was therefore assigned to two committees (consisting of reviewers, an Area Chair, and a Senior Area Chair) that reached independent decisions.  If both committees made the same recommendation, this recommendation was followed. If a single committee recommended acceptance, the paper was accepted (with the exception of a few cases in which the other committee identified what we considered a fatal flaw, e.g., an error in a key result).

This copy’s committee reached the following decision: **Reject**

The other committee assigned to the paper recommended **Accept (Poster)**.  You can find the other set of reviews, along with any follow up discussion with the authors here:
https://openreview.net/forum?id=xmX-WjAsf8y